# Multimodal MRI improves diagnostic accuracy and sensitivity to longitudinal change in amyotrophic lateral sclerosis

Pramod Kumar Pisharady [1✉], Lynn E. Eberly[2,1], Isaac M. Adanyeguh[1], Georgios Manousakis[3], Gaurav Guliani[3], David Walk [3,4] & Christophe Lenglet [1,4]

## Abstract

**Background** Recent advances in MRI acquisitions and image analysis have increased the utility of neuroimaging in understanding disease-related changes. In this work, we aim to demonstrate increased sensitivity to disease progression as well as improved diagnostic accuracy in Amyotrophic lateral sclerosis (ALS) with multimodal MRI of the brain and cervical spinal cord.

**Methods** We acquired diffusion MRI data from the brain and cervical cord, and T1 data from the brain, of 20 participants with ALS and 20 healthy control participants. Ten ALS and 14 control participants, and 11 ALS and 13 control participants were re-scanned at 6-month and 12-month follow-ups respectively. We estimated cross-sectional differences and longitudinal changes in diffusion metrics, cortical thickness, and fixel-based microstructure measures, i.e. fiber density and fiber cross-section.

**Results** We demonstrate improved disease diagnostic accuracy and sensitivity through multimodal analysis of brain and spinal cord metrics. The brain metrics also distinguished lower motor neuron-predominant ALS participants from control participants. Fiber density and cross-section provided the greatest sensitivity to longitudinal change. We demonstrate evidence of progression in a cohort of 11 participants with slowly progressive ALS, including in participants with very slow change in ALSFRS-R. More importantly, we demonstrate that longitudinal change is detectable at a six-month follow-up visit. We also report correlations between ALSFRS-R and the fiber density and cross-section metrics.

**Conclusions** Our findings suggest that multimodal MRI is useful in improving disease diagnosis, and fixel-based measures may serve as potential biomarkers of disease progression in ALS clinical trials.

## Plain Language Summary

ALS is a disease affecting the brain and spinal cord which leads to weakness and muscle wasting. It is important to be able to measure disease-related changes whilst clinical trials are ongoing to assess whether the treatments being tested are working. We imaged the brain and spinal cord of people with and without ALS at 3 time points over a year. We found changes in the brain and spine over time. This study demonstrates that brain imaging could be potentially used to assess changes in disease progression during clinical trials, giving an indication of whether the treatments being tested are having an effect.

[1] Center for Magnetic Resonance Research, Department of Radiology, University of Minnesota, Minneapolis, MN 55455, USA. [2] Division of Biostatistics, School of Public Health, University of Minnesota, Minneapolis, MN 55455, USA. [3] Department of Neurology, University of Minnesota, Minneapolis, MN 55455, USA. [4] These authors contributed equally: David Walk, Christophe Lenglet. ✉email: pramodkp@umn.edu

Diffusion MRI (dMRI) images the white matter structures indirectly by mapping the diffusion process of molecules, mainly water, in biological tissues[1]. The degeneration of axons affects the anisotropic characteristics of diffusion. Quantification of the changes in diffusion using dMRI represents the changes in anisotropic characteristics due to degeneration of axonal membranes, myelin sheaths, or the changes in axon density and coherence[2]. ALS, which is a fatal neurodegenerative disorder characterized by progressive weakness of the limb, bulbar and respiratory muscles[3], causes degeneration in motor and extramotor neural pathways[4]. Notably, the name of the disease itself references pathological changes in corticospinal tract (CST), a prominent white matter pathway for which the role of diffusion imaging has clear face validity. Quantification of the alterations in the central nervous system due to neurodegeneration has the potential to improve our understanding of ALS and to provide novel objective biomarkers of disease progression.

There exist several studies that report cross-sectional differences and longitudinal changes in brain dMRI metrics in ALS (see reviews in ref. [5–7]). Prior studies have shown the potential of diffusion tensor imaging (DTI) for early diagnosis of ALS before the development of clinical signs[8]. The degree of directionality of diffusion (anisotropic behavior) measured by Fractional Anisotropy (FA) is useful for assessing degeneration in tissue structure in patients without upper motor neuron (UMN) signs[9] but is not sufficiently sensitive at the single-patient level. FA abnormalities in the CST and callosal body have been shown to correlate with clinical UMN burden[10]. Both FA and tractography based fiber connectivity measures have also been shown to provide complementary information, with FA sensitive to the detection of patients vs. control group differences and connectivity measures correlating with disease progression rate[11]. Tract-based spatial statistics (TBSS) has also been used for ALS data analysis[12–14]. Li et al.[5] provided a meta-analysis of DTI based studies in ALS covering the developments from 1990 to 2010. They reported consistent reduction in FA in the bilateral frontal white matter, cingulate gyrus and the posterior limb of bilateral internal capsules. Reduced FA in the cingulate gyrus and superior longitudinal fasciculus, areas that play a crucial role in emotional processing, motivation, and goal-directed behaviors, has also been reported[15,16], which may provide a potential imaging correlate for the cognitive and behavioral symptoms commonly seen in ALS.

Mendili et al. provided a comprehensive review of spinal cord imaging methods, their advantages, and drawbacks[17]. They discussed magnetization transfer imaging, MR spectroscopy, and functional MRI in addition to dMRI methods. We have provided a detailed overview of spinal cord dMRI studies[18–23] in ALS in our prior work[24]. In our spinal cord MRI paper, we reported the results of a tract-specific and along-the-tract analysis of different spinal cord measures including FA and cross-sectional area (CSA), from C2 to C6. We demonstrated that the tract-specific analysis with segmentation of ascending and descending tracts in the spinal cord white matter substantially increases the sensitivity of dMRI to disease-related changes in ALS. We identified the tracts and spinal levels affected in ALS, and reported the involvement of sensory pathways in ALS. We also noted strong correlations between spinal diffusion and cross-sectional area metrics and the spinal components of ALSFRS-R. In this study, our focus is on multimodal analysis of brain and spinal cord measures.

van der Burgh et al. conducted a multimodal longitudinal study of disease-related structural changes in the brain in ALS, using dMRI and T2 imaging[25]. They assessed cortical thickness, subcortical volume, and white matter connectivity, and reported distinct patterns of cerebral degeneration based on heterogeneity of phenotype and C9orf72 genotype. They reported widespread gray and white matter involvement at baseline, and extensive loss of white matter integrity over time in patients with a C9orf72 mutation. In C9orf72-negative patients, cortical thinning of motor and frontotemporal regions and loss of integrity of white matter pathways associated with the motor cortex were noted. They also reported more white matter involvement at baseline in spinal onset participants and greater gray matter involvement in bulbar onset participants. Querin et al. conducted a multimodal analysis of spinal cord data, with dMRI, T2, and magnetization transfer ratio (MTR) imaging, with a goal to improve diagnostic performance[26]. They reported a sensitivity of 88% and specificity of 85% in distinguishing participants with ALS from controls. Borsodi et al. compared DTI metrics, MTR, and the effective transversal relaxation rate R2* from the brain to investigate their potential correlation with clinical variables[27]. They concluded that DTI revealed more widespread white matter tissue changes than MTR and R2*.

These multimodal studies used either brain or spinal cord data, not both, and primarily used tract-based DTI methods that do not resolve crossing fiber bundles. In this study, we compare multiple measures from the brain and spinal cord in cohorts of people with ALS and healthy control participants. We use the *ball-and-stick* multi-compartment model that resolves crossing fibers (FSL *bedpostx* algorithm[28]) in a TBSS analysis rather than using the DTI model. The *ball-and-stick* model uses a ball compartment to represent the isotropic diffusion and multiple stick components to represent the anisotropic diffusion in crossing fibers. We present results from multiple MRI-derived measures of the brain including cortical thickness, FA, sum of fiber volume fractions (Fsum), and fixel-based measures, comparing ALS participants with control participants. The *fixel* refers to a single fiber population within a voxel, and the fixel-based analysis (FBA) conducts statistical analysis of the fixel-based measures, the fiber density (FD), the fibre-bundle cross-section (FC), and the combined measure of fiber density and cross-section (FDC)[29]. We combine these measures from the brain with FA and CSA of the spinal cord. We hypothesize that combining diffusion and morphometry measures from the brain and spinal cord in a multimodal analysis will improve the discrimination power between ALS and control participants. Our results demonstrate that a multimodal analysis of brain and spinal cord MRI data can provide increased diagnostic accuracy and sensitivity, and the fixel-based measures are useful in measuring disease progression over shorter durations. We also noted statistically significant correlations between fixel-based measures and the ALSFRS-R.

## Methods

**Study participants and design.** We recruited participants who met revised El Escorial Criteria[30] for clinically possible, probable, or definite ALS from the ALS Association Certified Treatment Centers of Excellence at the University of Minnesota and Hennepin County Medical Center. Healthy control participants with matching age range and sex frequency were recruited from the general public. Participants who had neurologic illnesses other than ALS, the inability to tolerate MRI scanning, or who failed to meet MRI safety criteria were all excluded from the study. Exclusion criteria for control participants included (1) presence of neurological illness, (2) abnormal neurological examination, or (3) abnormal cognitive screening. Written informed consent was collected using procedures approved by the Institutional Review Board: Human Participants Committee of the University of Minnesota. All participants underwent imaging at time of enrollment, in compliance with all the ethical regulations. The participants were asked to return for follow-up visits at 6 and 12 months after enrollment. We previously reported spinal cord MRI findings[24] and brain magnetic resonance spectroscopy findings in this cohort[31,32].

**Clinical assessments**. Neuromuscular examination was done by a neuromuscular neurologist (D.W., G.M., G.G.) at enrollment and at the 12-month follow-up. ALS Functional Rating Scale-Revised (ALSFRS-R)[33] score was calculated to measure the functional impairment in ALS participants at enrollment, six-month, and 12-month visits. Clinical staging was done using an algorithm developed by King's College London (stage 1 mild – stage 4 advanced)[34]. Behavioral and cognitive status was assessed in all participants at all visits using the Edinburgh Cognitive Behavioral ALS Screen (ECAS)[35]. Total ECAS and its ALS-specific component scores were recorded. UMN burden score was derived based on the neuromuscular examination as previously described in ref. [31] and ranges from 0 to 6, with a higher score indicating greater UMN burden. A lower motor neuron (LMN) burden score was derived based on the scoring system proposed by Devine et al.[36]. The LMN burden score ranges from 0 to 12, with a higher score indicating greater LMN burden. All clinical assessments were done within one week of the MRI exam. Current riluzole use was also documented. The disease duration was calculated as the time from the date of first reported symptoms to the date of the MRI exam.

**MRI data collection**. Diffusion MRI data were acquired using 3 T Siemens Trio and Prisma scanners at the Center for Magnetic Resonance Research (CMRR) at the University of Minnesota. Brain data were acquired using diffusion encoding along 128 directions with a $b$-value of 1500 s/mm$^2$. Seventeen additional volumes without diffusion encoding were equally interleaved in the dataset yielding a total of 145 volumes. We obtained 90 slices with thickness 1.8 mm and voxel size $1.81 \times 1.81$ mm$^2$ (FoV = $106 \times 106$). Two sets of data were collected during each session, with reversed phase encoding directions (anterior to posterior and posterior to anterior). They were subsequently combined to correct for distortions and to increase signal to noise ratio[37]. T1 images are acquired at an isometric resolution of 1 mm with a size of 176x256x224 voxels. The protocol used for spinal cord data acquisition was described in our prior work[24].

**MRI data processing**. Supplementary Fig. 1 summarizes the data processing and multimodal statistical analysis steps. The data were corrected for distortions due to eddy currents, susceptibility-induced off-resonance artifacts and subject motion[37,38]. Brain was extracted from both diffusion and T1 images. Both ball-and-stick (*bedpostx*) and DTI (*dtifit*) models were subsequently fitted to the corrected diffusion data using FSL[38]. DTI metrics as well as the sum of fiber volume fraction (Fsum) maps were calculated for further statistical analysis. We included Fsum in the analysis as Fsum is considered an extension of FA, with fiber crossings taken into account. Cortical thickness was calculated using Freesurfer software[39,40]. The fixel-based metrics (see the next subsection) were calculated using Mrtrix software[29]. For the multimodal analysis we also used FA and CSA measures from the spinal cord, the extraction of these measures has been reported previously[24].

**Fixel-based analysis**. The fixel-based analysis (FBA) addresses the challenge of resolving crossing fibers by extracting measures that are linked to individual fiber populations (called *fixels*) within a voxel that are directly related to the white matter anatomy. The amplitudes of the fiber orientation distributions (FOD) were calculated to derive the fixels, using constrained spherical deconvolution[41]. The FODs were segmented to evaluate the orientation and number of fixels in each voxel and generate the fiber density (FD), fiber-bundle cross-section (FC) and a product of both, fiber density and cross-section (FDC) maps. The identification of related fixels is driven by a template tractogram

generated with anatomically relevant white matter tracts[42]. FBA metrics describe the total intra-axonal volume that is related to the white matter microstructure; and the fiber bundle cross-section that describes the macroscopic differences in fiber bundle. Due to their specificity, FBA metrics are able to differentiate between affected and non-affected tracts within a given voxel unlike the conventional DTI that resolves only one fiber population[43]. Therefore, FBA is more useful in characterizing the complex geometry of the white matter with more clinical relevance pertaining to its biological specificity.

**Tests to study the effect of scanner on MRI metrics**. Due to a scanner upgrade during the study, we used both Trio and Prisma scanners for both baseline and follow-up scans. Sequence parameters and protocols were carefully matched. At baseline, 10 of the 20 ALS participants were scanned on Trio and the other 10 were scanned on Prisma. For the controls, 7 participants were scanned on Trio and 13 participants were scanned on Prisma at baseline. We conducted two-sample, two-tailed, unpaired t-tests between the two scanner groups in the control participants at baseline, and did not find statistically significant differences in any of the metrics we used. We also conducted a similar analysis between the two scanner groups in ALS participants, which too did not detect statistically significant differences.

**Statistical analysis of single modality data**. We first conducted statistical analysis of FA data, as well as the sum of fiber volume fractions (Fsum), from the cross-sectional and longitudinal data, using whole brain tract-based spatial statistics (TBSS)[44]. Mean cortical thickness measurements of the baseline ALS and control participants were compared using two-sample, two-tailed, unpaired t-tests across the corresponding left and right ROIs. Cortical thickness measurements at baseline and follow-up (6 and 12 months) for ALS participants were also analyzed for longitudinal change, using two-tailed paired t-test. Cross-sectional and longitudinal analysis of FD, FC, and FDC metrics were done using whole brain fixel-based analysis using non-parametric permutation testing[45]. The affected regions, their cluster size volume, minimal p-values, and the MNI co-ordinates of the center-of-gravity of these regions are extracted using the respective atlases for the white matter (JHU ICBM 1 mm[46]), cortex (HarvardOxford cort maxprob thr25 1 mm[47]), and subcortex (HarvardOxford sub maxprob thr25 1 mm[48]).

**Statistical analysis of multi-modal data**. In order to extract brain and spine metrics for the multimodal analysis, we used the following strategy. For brain FA and Fsum, we created an ROI mask of the voxels with statistically significant group difference between ALS and control participants (in the case of cross-sectional analysis) or between time points for ALS participants (in the case of longitudinal analysis), as detected by the TBSS analysis. Mean values for the metrics of interest were extracted, within this mask, and were used for subsequent analyses. We followed a similar procedure for the fixel-based metrics. We note however that this was only performed for the longitudinal data, as we did not find a statistically significant difference for those metrics in the cross-sectional analysis (see the Results section). The cortical thickness measure used for cross-sectional multimodal analysis is the cortical thickness of the precentral area and that for longitudinal multimodal analysis is the cortical thickness of the superior temporal area (average of left and right ROIs in both cases). We chose these areas for the multimodal analysis as we found cortical thinning in these areas to be most significant in the corresponding cross-sectional and longitudinal single modal

analyses. The measures used from the spinal cord are the mean CSA at C2 level and the mean FA in CST at the C2 level.

The cross-sectional multimodal analysis was performed with two measures from the brain (Fsum or FA and cortical thickness) and two measures from the spinal cord (CSA and CST FA, both at the C2 level). The p-values reported from the unimodal analysis were corrected for multiple testing (across the multiple modalities) using the Bonferroni-Holm method[49]. The multimodal analysis was done using multivariable logistic regression with the disease status (patient or control) as the response variable and the above four metrics as predictors. Leave-one-out cross-validation was done and the mean accuracy, sensitivity, and specificity are reported. In the longitudinal case we included the FDC measure from brain as well, thereby using three measures from the brain (Fsum or FA, cortical thickness, and FDC) and two measures from the spinal cord (CSA and CST RD, both at the C2 level). The predictor variable in the longitudinal case is the difference between the metrics at baseline and 12-month follow-up.

**Effect size calculations**. For the cross-sectional data, the effect size is estimated using Cohen's $d$ and is calculated as the difference of the group means M1 and M2 divided by the root mean square value of the respective standard deviations SD1 and SD2.

$$Cohen's\ d = (M_1 - M_2)/SD_{rms} \quad (1)$$

$$SD_{rms} = \sqrt{((SD_1^2 + SD_2^2)/2)} \quad (2)$$

For the longitudinal data the effect size is represented using standardized response mean (SRM) and is calculated as the ratio of mean change D and the standard deviation of the change SD.

$$SRM = D/SD \quad (3)$$

**Correlations with functional status**. We studied the correlations of the imaging metrics with ALSFRS-R of the 20 ALS participants at the initial visit. The correlation is estimated by calculating the Pearson correlation coefficient between ALSFRS-R and mean FDC in left and right CST. We also studied correlations between the within-person change in these metrics and the within-person change in ALSFRS-R over a one-year period.

**Statistics and Reproducibility**. The statistical analysis was conducted as described in the previous subsections. Sample size and other statistical parameters are detailed in Table 1.

**Reporting summary**. Further information on research design is available in the Nature Portfolio Reporting Summary linked to this article.

# Results

**Cohort characteristics**. Twenty ALS and 20 control participants were scanned. We have reported the demographic and clinical features of this cohort in our prior publication which focused on spinal cord data[24]. The table presenting demographic and clinical features, updated with 6-month data, is included here (Table 1). A diagram showing the flow of participants is shown in Supplementary Fig. 2. The ALS cohort was an early-stage cohort on average (mean ALSFRS-R of 40.0, 60% participants in King's Stage 1 or 2 at time of enrollment). No ALS participants had a documented history of ALS in first-degree relatives. Three had negative genetic testing using an ALS gene panel, two tested negative for C9orf72 repeat expansions. The others did not undergo genetic testing. One control participant scored (ECAS

score 103) slightly below the cognitive screen normal cutoff (ECAS score 105) at enrollment but was included as this was determined to be due to a misunderstanding of one of the instructions; cognitive scores at 6 and 12 months were well within normal limits for this participant (ECAS scores of 125 and 123, respectively).

Repeat scanning at 6-month follow-up was done for 10 ALS and 14 control participants, and the repeat scanning at 12-month follow-up was done for 11 ALS and 13 control participants. The ALSFRS-R of ALS participants who were re-scanned at 12 months decreased by a mean of 5.0 points, or an average slope of −0.42 points/month, at the one-year follow-up. None of the participants in King's stages 3 or 4 at enrollment returned for the one-year follow-up. Four deaths occurred in the ALS cohort and no deaths occurred in the control cohort, between the baseline and one-year follow-up.

**Cross-sectional analysis results**
*Cross-sectional data: TBSS results with sum of fiber volume fractions and FA*. Figure 1 shows results of the TBSS analysis (axial slices) for the ALS and control participants. The ALS group exhibited lower fiber volume fractions (Fsum, panels in rows one and two) compared to the healthy control group in many regions. Clusters with significantly lower Fsum included the corpus callosum (regions of genu, body and splenium), corticospinal tract, posterior limb of the internal capsule, retrolenticular part of internal capsule, external capsule, anterior, superior and posterior corona radiata, posterior thalamic radiation, superior longitudinal fasciculus, tapetum, sagittal stratum, cerebellar peduncle, superior cerebellar peduncle, superior fronto-occipital fasciculus, anterior limb of internal capsule, uncinate fasciculus (bilateral findings for all the regions of interest, ROIs), and the pontine crossing tract (refer Supplementary Data 1 for the full list of regions with cluster size and MNI coordinates, where lower Fsum is detected). FA was also lower in many of these regions (lower FA in ALS participants, panels in rows three and four), but not the left superior fronto-occipital fasciculus, left anterior limb of internal capsule, and the right uncinate fasciculus. Both precentral gyrus (which includes the primary motor cortex) and post central gyrus (which includes the primary somatosensory cortex) show detectable group differences by Fsum as well as FA. The Fsum analysis detected a higher number of voxels with significant differences between the ALS and control cohorts in all the regions (Fig. 1a). Other cortical areas with significant differences in Fsum but not FA include the supplementary motor cortex (the juxtapositional lobule cortex), precuneus cortex, middle temporal gyrus, superior frontal gyrus, and the frontal pole. Note that our observations about these cortical areas are based on dMRI metrics tested in the white matter adjacent to those cortical areas. In all the above analyses, Fsum and FA were lower in the ALS cohort in all the regions detected. There were no regions in which Fsum or FA was higher in the ALS cohort compared to controls.

*Cross-sectional data: Cortical thickness*. Statistically significant cortical thinning in the ALS participants is noted in the precentral cortex and pars opercularis (Bonferroni-Holm corrected p-values 0.004 and 0.017 respectively). We did Bonferroni-Holm correction[49] for multiple testing for the 34 cortical regions we tested as per the Desikan–Killiany atlas[50]. We noted marginally significant cortical thinning in many other areas, but those p-values did not survive the correction for multiple testing. The areas where we found marginally significant cortical thinning are (uncorrected p-value from two-sample t-test in parenthesis): paracentral (0.009), postcentral (0.037), caudal middle frontal (0.037), superior parietal (0.023), supramarginal (0.030), and

**Table 1 Demographic and clinical features of ALS and control cohorts at initial visit and at follow-up.**

| | Visit 1 (baseline) | | Visit 2 (6-month) | | Visit 3 (12-month) | |
|---|---|---|---|---|---|---|
| | ALS | Control | ALS | Control | ALS | Control |
| Sample size *n* | 20 | 20 | 10 | 14 | 11* | 13 |
| Sex ratio, male:female | 11:9 | 10:10 | 5:5 | 6:8 | 7:4 | 6:7 |
| Age, years | 57.5 ± 9.8 (31–74) | 57.0 ± 8.7 (30–69) | 58.7 ± 11.2 (31–70) | 59.0 ± 6.1 (48–68) | 57.8 ± 11.2 (32–71) | 57.0 ± 9.6 (31–69) |
| Riluzole use, yes:no | 9:11 | -- | 4:6 | -- | 6:5 | -- |
| Disease duration, months | 38.5 ± 42.5 (3.4–147.7) | -- | 49.7 ± 55.0 (9.4–153.7) | -- | 50.1 ± 53.8 (15.8–160.0) | -- |
| Site of onset | Upper limb (10) Lower limb (5) Bulbar (5) | -- | Upper limb (5) Lower limb (3) Bulbar (2) | -- | Upper limb (4) Lower limb (5) Bulbar: (2) | -- |
| King's disease stage (number of participants) | Stage 1 (5) Stage 2 (7) Stage 3 (7) Stage 4 (1) | -- | Stage 1 (5) Stage 2 (3) Stage 3 (1) Stage 4 (1) | -- | Stage 1 (2) Stage 2 (2) Stage 3 (2) Stage 4 (5) | -- |
| El Escorial classification (number of participants) | Possible (7) Probable (8) Definite (5) | -- | -- | -- | Possible (3) Probable (5) Definite (3) | -- |
| ALSFRS-R total score (0 most severe – 48 normal) | 40.0 ± 5.5 (27–45) | -- | 40.6 ± 4.1 (33–47) | -- | 38.0 ± 4.6 (31–45) | -- |
| ALSFRS-R spinal sub-score (Q4-Q12) (0 most severe – 36 normal) | 29.7 ± 5.3 (15–36) | -- | 30.6 ± 4.4 (24–36) | -- | 28.6 ± 4.3 (22–36) | -- |
| ECAS total score (0 most severe – 136 normal) | 113.3 ± 7.9 (90–126) | 119.8 ± 8.7 (103–129) | 116.3 ± 8.4 (106–135) | 123.0 ± 7.1 (107–132) | 116.9 ± 7.2 (106–129) | 122.4 ± 7.2 (108–134) |
| ECAS ALS specific sub-score (0 most severe – 100 normal) | 84.4 ± 7.6 (59–95) | -- | 86.6 ± 6.9 (77–100) | -- | 86.2 ± 5.5 (79–97) | -- |
| UMN burden score (0 normal – 6 most severe) | 2.5 ± 1.3 (1–6) | -- | -- | -- | 2 ± 1.5 (0–4) | -- |

*Four deaths occurred in ALS participants before the 12-month visit; there were no deaths in controls. The 5 remaining ALS participants lost to follow-up were male:female 2:3 and age 58 ± 9.3 years; the 7 controls not rescanned were male:female 4:3 and age 58.7 ± 7.1 (ALS Amyotrophic lateral sclerosis, ALSFRS-R - ALS functional rating scale - revised, ECAS Edinburgh Cognitive Behavioral ALS Screen, UMN Upper motor neuron).

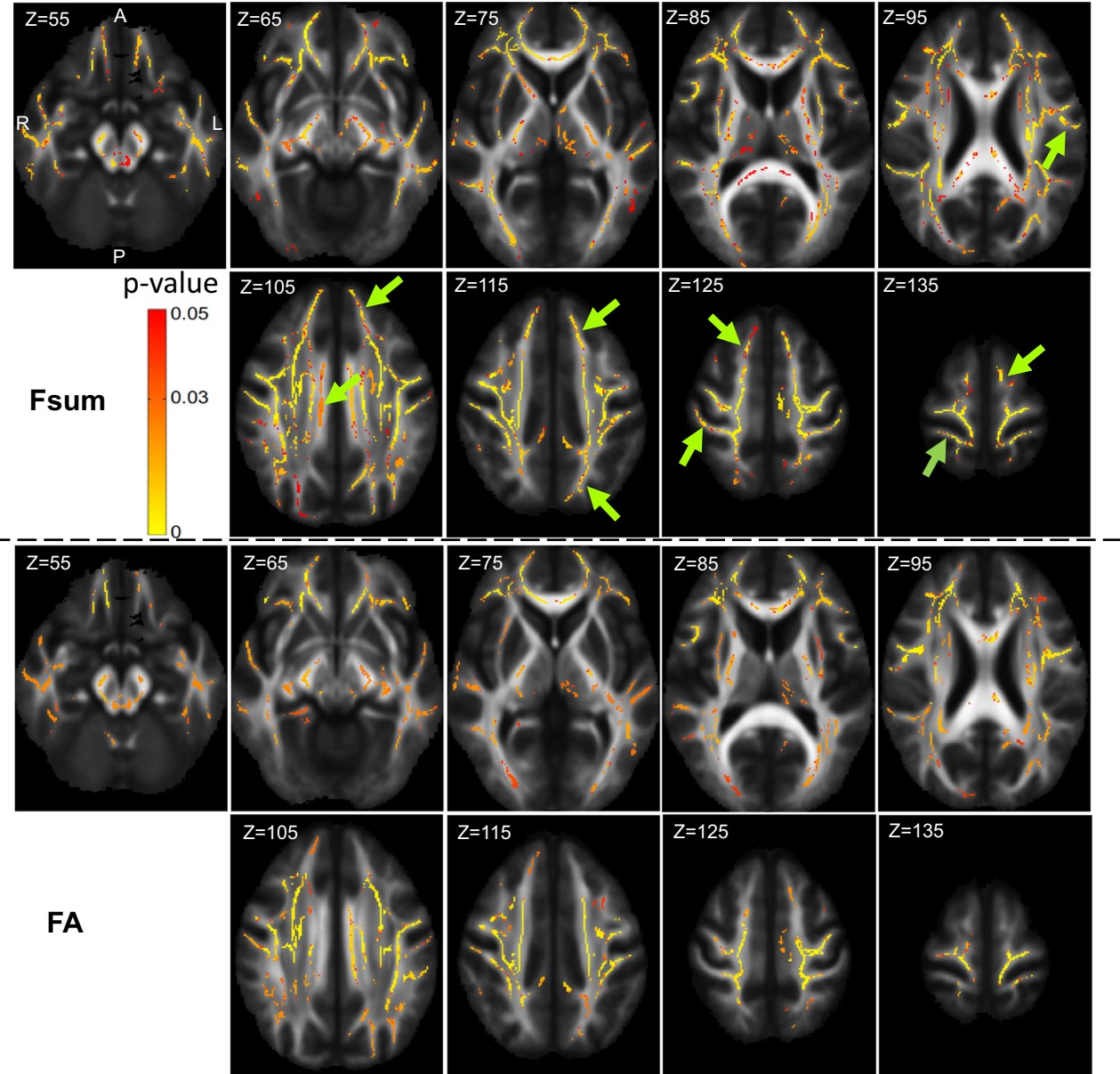

**Fig. 1 Whole brain cross-sectional group difference (ALS vs. control) - Tract based spatial statistics.** Cross-sectional TBSS results at initial visit with comparison of group differences ($n = 20$ in each group) in Fsum (panels in row 1 and 2) and FA (panels in row 3 and 4). Color-coded regions show the areas where statistically significant group differences exist ($p < 0.05$, corrected for multiple comparisons across space). Green arrows indicate regions where group differences are detected by Fsum but not by FA (Fsum - Sum of fiber volume fractions, FA Fractional anisotropy).

inferior parietal cortices (0.011). Supplementary Fig. 3 denotes these regions with a letter C in parentheses.

*Cross-sectional data: Brain fiber density and fiber cross-section.* We did not find significant cross-sectional differences between ALS and control participants in FD, FC, or FDC at baseline.

*Cross-sectional data: Multimodal analysis of brain and spine data.* We have previously reported DTI metrics of individual spinal cord tracts and the CSA of the spinal cord (white matter, gray matter, and the whole cord) from C2 to C6[24]. Here we report the mean CSA of the whole cord and the mean FA of CST at the C2 level, and use those for a multimodal analysis with metrics from the brain.

In the cross-sectional multimodal analysis we used four independent metrics, two diffusion metrics (one each from the brain and the spine), and two morphology metrics (one each

from the brain and the spine). The metrics used are mean FA extracted from the voxels from TBSS analysis where we found statistically significant group differences between ALS and control participants (which cover affected regions listed in the TBSS results section, including corticospinal tract and corpus callosum), cortical thickness of the precentral area, FA in CST at C2 level, and CSA at C2 level. We first present the unimodal analysis results with these metrics as well as with Fsum (which is extracted in a similar manner as FA). Group differences in these diffusion and morphometric measures, as well as the effect sizes and p-values from two-sample t-tests, are shown in Fig. 2 and Table 2, respectively. Brain FA provided the greatest effect size and lowest p-value, followed by brain Fsum. In Table 3 we show the group classification performance of individual metrics as well as multimodal metric. The multimodal metric with brain FA, precentral cortical thickness, CSA at C2, and FA in CST at C2 provided the highest accuracy (87.5%), sensitivity (90%), and

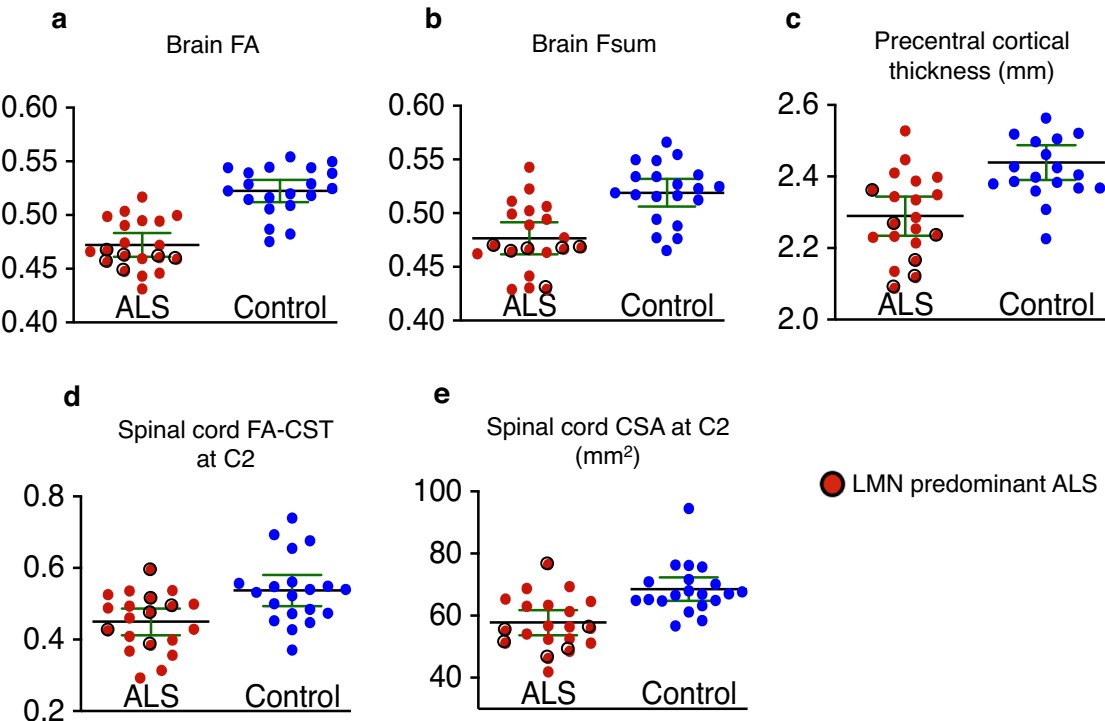

**Fig. 2 Summary of cross-sectional group difference (ALS vs. control) in different brain and spinal cord measures. a–e** Scatter dot plots of multiple measures from the brain and the spinal cord showing the group difference between ALS participants ($n = 20$) and controls ($n = 20$). For each group, the plot shows the mean (blackline) with 95% confidence interval (green lines) (ALS Amyotrophic lateral sclerosis, FA Fractional anisotropy, Fsum Sum of fiber volume fractions, CST Corticospinal tract, C2 Spinal cord C2 level, CSA Cross-sectional area).

**Table 2 Effect size and p-values for group comparisons of individual measures: cross-sectional data.**

| Measure (mean FA and Fsum of the statistically significant voxels in the TBSS analysis, cortical thickness of the precentral area, CSA at C2, and FA in CST at C2) | Effect size (Cohen's d) | P-value: two-sample t-test for group difference | Corrected p-value (Bonferroni-Holm) | Low UMN score (< 3) ALS participants vs. controls (corrected p-value) | LMN predominant ALS vs. controls (corrected p-value) |
|---|---|---|---|---|---|
| Brain FA | 2.20 | $3 \times 10^{-8*}$ | $1.5 \times 10^{-7*}$ | $1.4 \times 10^{-3*}$ | $1.4 \times 10^{-8*}$ |
| Brain Fsum | 1.43 | $6 \times 10^{-5*}$ | $2.4 \times 10^{-4*}$ | $6.5 \times 10^{-2}$ | $2.5 \times 10^{-5*}$ |
| Precentral cortical thickness | 1.35 | $1.3 \times 10^{-4*}$ | $3.9 \times 10^{-4*}$ | $4.4 \times 10^{-2*}$ | $3.3 \times 10^{-3*}$ |
| Spinal cord CST FA at C2 | 1.00 | $3.2 \times 10^{-3*}$ | $3.2 \times 10^{-3*}$ | $4.4 \times 10^{-2*}$ | $1.7 \times 10^{-1}$ |
| Spinal cord CSA at C2 | 1.29 | $2.3 \times 10^{-4*}$ | $4.6 \times 10^{-4*}$ | $9.1 \times 10^{-3*}$ | $6.4 \times 10^{-2}$ |

*$p$-value < 0.05 (ALS Amyotrophic lateral sclerosis, FA Fractional anisotropy, Fsum Sum of fiber volume fractions, CST Corticospinal tract, C2 Spinal cord C2 level, CSA Cross-sectional area, UMN Upper motor neuron, LMN Upper motor neuron).

**Table 3 Group classification performance with separate metrics and with multimodal metrics for the cross-sectional data.**

| Measure | All ALS participants vs. controls | | | Low UMN score (< 3) ALS participants vs. controls | | |
|---|---|---|---|---|---|---|
| | Accuracy (%) | Sensitivity (%) | Specificity (%) | Accuracy (%) | Sensitivity (%) | Specificity (%) |
| Brain FA | 82.5 | 80.0 | 85.0 | 75.9 | 69.2 | 85.0 |
| Brain Fsum | 72.5 | 70.0 | 75.0 | 67.0 | 61.5 | 80.0 |
| Precentral cortical thickness | 72.5 | 70.0 | 75.0 | 67.0 | 46.2 | 90.0 |
| Spinal cord CST FA at C2 | 60.0 | 55.0 | 65.0 | 62.1 | 46.2 | 80.0 |
| Spinal cord CSA at C2 | 72.5 | 65.0 | 80.0 | 75.9 | 69.2 | 85.0 |
| **Multimodal*** | **87.5** | **90.0** | **85.0** | **93.1** | **92.3** | **95.0** |

*Multimodal analysis, which provided the best accuracy and sensitivity, included brain FA, cortical thickness, cord CST FA at C2, and cord CSA at C2 (ALS Amyotrophic lateral sclerosis, FA Fractional anisotropy, Fsum Sum of fiber volume fractions, CST Corticospinal tract, C2 Spinal cord C2 level, CSA Cross-sectional area, UMN Upper motor neuron).

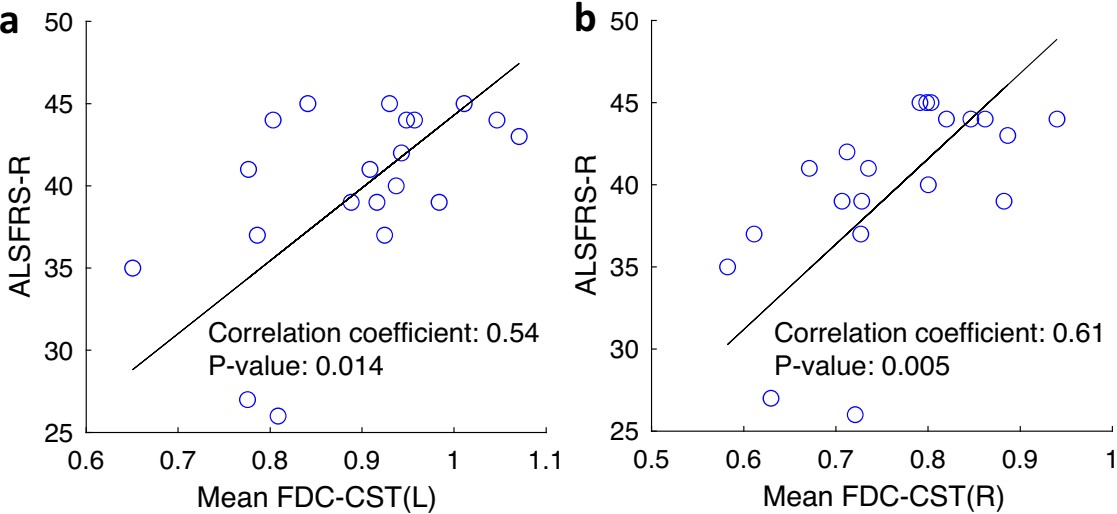

**Fig. 3 Correlation between ALSFRS-R and brain fixel-based measures.** Correlation between ALSFRS-R and (**a**) mean FDC of brain CST left and (**b**) mean FDC of brain CST right. Sample size n = 20 (ALSFRS-R ALS functional rating scale - revised, FDC Fiber density and cross-section, CST Corticospinal tract).

specificity (85%) in predicting disease status compared to the metrics provided by individual measures. Although Fsum detected more affected regions in the TBSS analysis, FA provided a higher classification rate (see Discussion section for a discussion on this), hence Fsum was not included in the multimodal analysis.

*Cross-sectional data: Distinguishing participants with low UMN burden from controls.* We also studied the ability of the multimodal metrics to distinguish ALS participants with low UMN burden from controls. We analyzed the data from those 13 ALS participants who had a UMN burden score of 1 or 2 (out of 6), and found that all metrics except Fsum (which did not survive correction for multiple testing) differentiated these participants from controls, albeit with lower significance (i.e., higher p-value) compared to the whole group analysis (Table 2). We did not find statistically significant differences in any of these metrics between low and high UMN burden score participants. We also report the accuracy, sensitivity, and specificity of the separate and multimodal metrics in distinguishing these low UMN score participants from controls. The multimodal analysis provided much higher accuracy and sensitivity compared to the corresponding single modality metrics in this case. The multimodal analysis with brain FA, precentral cortical thickness, CSA at C2, and FA in CST at C2 provided an accuracy of 89.66%, sensitivity of 89%, and specificity of 90% in predicting the disease status of these participants (Table 3).

*Cross-sectional data: Distinguishing LMN predominant ALS participants from controls.* We have defined the LMN predominant participants as participants who received low UMN score (1 or 2) and high LMN score. High LMN score is defined as severe weakness (antigravity or worse) in 2 or more limbs. Six participants were identified in this subgroup (Fig. 2). Brain FA, Fsum, and precentral cortical thickness distinguished these participants from controls with high statistical significance (Table 2, last column). However the spinal cord measures did not distinguish the LMN predominant ALS participants from controls. As we only had 6 participants in this category, we have not conducted the accuracy and sensitivity cross-validation analysis for this subgroup.

*Cross-sectional data: Analysis based on site of onset.* We performed an exploratory analysis comparing ALS participants with upper-limb onset (10 participants), lower-limb onset (5 participants), or bulbar-onset ALS (5 participants). Limb onset participants had lower Fsum and FA of the brain, cortical thickness, and FA and CSA of the cervical cord than bulbar onset participants. Lower limb onset participants had lower cortical thickness (p-value 0.05) than upper limb onset participants. For the other measures (Fsum and FA of the brain, CSA and FA of the cord), the value was slightly lower in the upper limb onset participants, but this difference was not statistically significant.

*Cross-sectional data: Analysis based on King's staging.* We conducted an exploratory analysis comparing the MRI measures of participants who were at King's stage 3 or 4 at baseline with those measures of participants at King's stage 1 or 2. We found marginally significant (that is p-values were not significant after correction for multiple testing) differences in Fsum and FA between these participants. Both Fsum (uncorrected p-value: 0.022) and FA (uncorrected p-value: 0.025) were lower in participants at King's stage 3 or 4 compared to participants at King's stage 1 or 2.

*Cross-sectional data: Correlations with functional status.* We did not find statistically significant correlations between ALSFRS-R and FA, Fsum, or cortical thickness. However we found strong correlations between ALSFRS-R and mean FDC of both left (Fig. 3a) and right brain CST (Fig. 3b). There were no statistically significant correlations between any of the MRI measures and UMN burden score.

In our prior work[24] we found that gray matter FA at C2 at the initial visit predicted study withdrawal due to progression of functional deficits from ALS. By contrast, brain metrics at enrollment did not predict survival or study withdrawal due to disease progression.

**Longitudinal analysis results**
*Longitudinal data: TBSS results with sum of fiber volume fractions.* Figure 4 (upper two rows) shows the longitudinal TBSS analysis for the ALS participants (baseline vs. 12 months). Longitudinal decreases in both Fsum and FA were detected in the cerebral peduncle, posterior limb of internal capsule, retrolenticular part of internal capsule (bilateral findings for all the ROIs), right thalamus, left pallidum, and brain-stem (refer Supplementary Data 1 for the full list of regions with cluster size and MNI

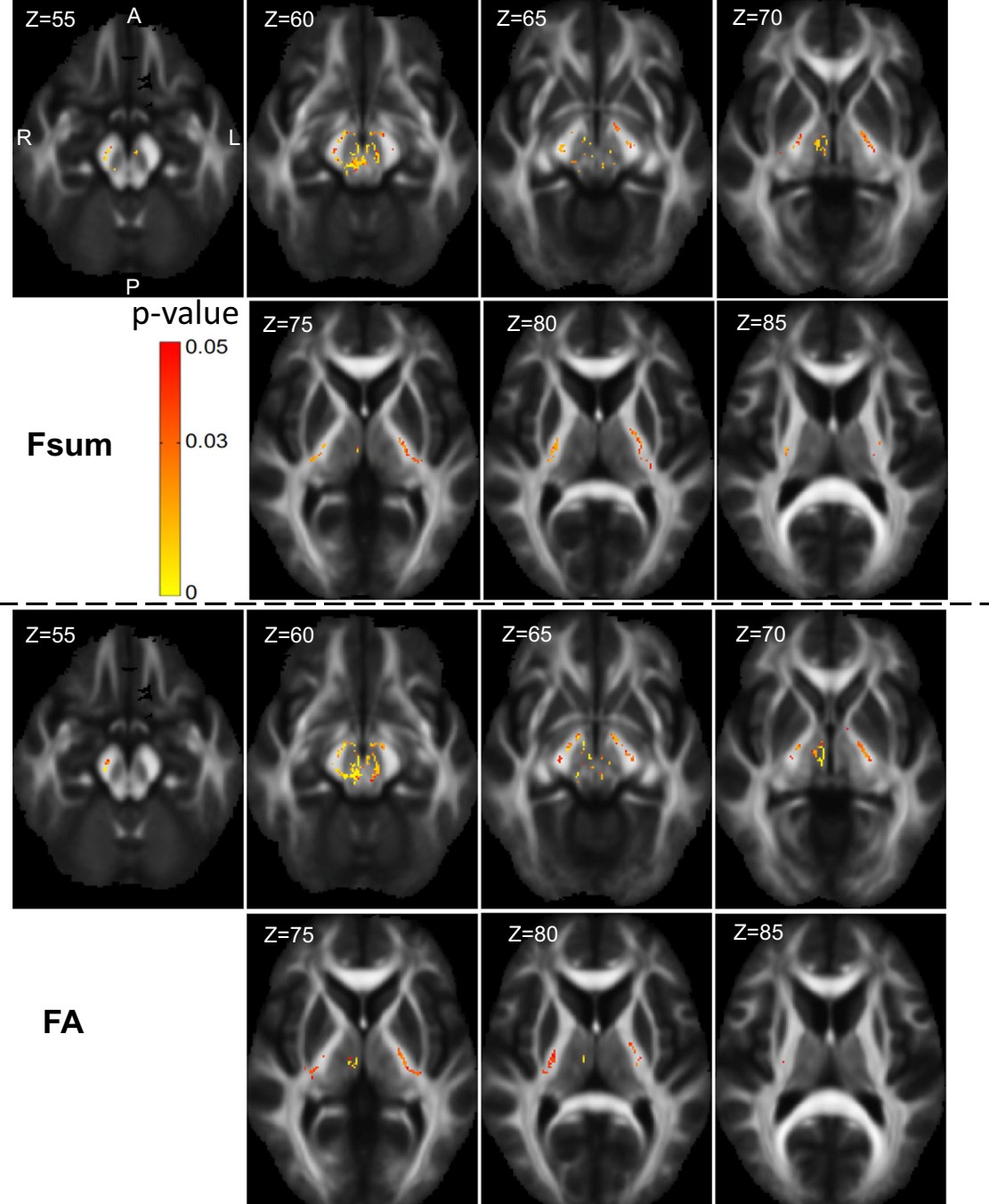

**Fig. 4 Whole brain longitudinal change in ALS participants - Tract based spatial statistics.** Longitudinal TBSS results showing regions where longitudinal changes are noted in the 12-month follow-up ALS data compared to the baseline ALS data ($n = 11$), in Fsum (panels in row 1 and 2) and FA (panels in row 3 and 4). Color-coded regions show the areas where statistically significant group differences exist ($p < 0.05$, corrected for multiple comparisons across space). (Fsum Sum of fiber volume fractions, FA Fractional anisotropy).

coordinates, where decrease in Fsum is detected). We did not find any longitudinal change in the control participants.

*Longitudinal data: Cortical thickness.* Supplementary Fig. 4 shows longitudinal change in cortical thickness at the 12-month follow-up in individual ALS participants ($n = 11$). After a Bonferroni-Holm correction for 34 regions tested, we noted only marginally significant group reduction (that is, those p-values did not survive the correction for multiple testing) in cortical thickness in the ALS cohort in the following regions (uncorrected p-value from paired t-test in parentheses): medial orbital frontal (0.041),

temporal pole (0.027), pars opercularis (0.018), superior temporal (0.018), and the middle temporal cortices (0.022). Supplementary Fig. 3 shows these regions with letter L in parentheses. We did not find any longitudinal change in the control participants.

*Longitudinal data: Fiber density and Fiber cross-section.* FDC, FD, and FC detected greater longitudinal decline than FA or Fsum. More regions showed decline in FDC, compared to FD or FC. Areas of statistically significant longitudinal change in the ALS cohort are shown in Fig. 5. Decreased FDC was detected in corticospinal tract, superior corona radiata, cerebral peduncle,

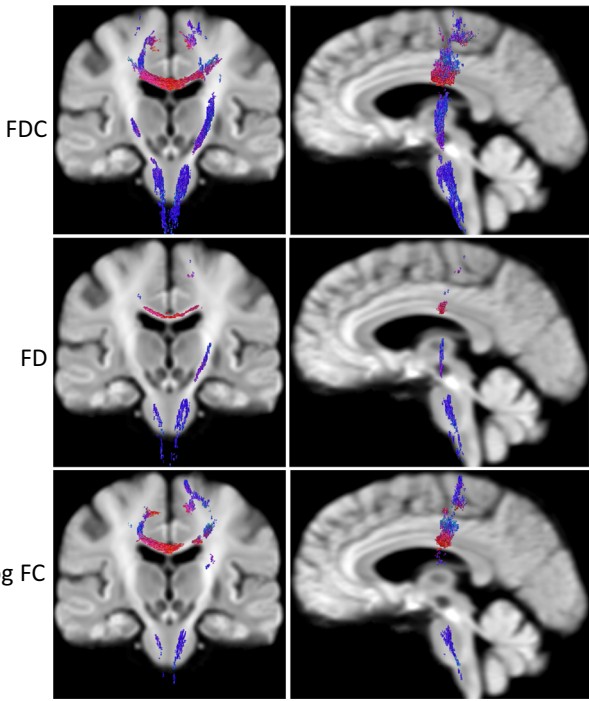

**Fig. 5 Whole brain longitudinal change in ALS participants - Fixel Based Analysis.** Figure 2 Longitudinal FBA results showing coronal and sagittal views of fiber alterations. Decrease in FDC (row 1), FD (row 2), and log FC (row 3) are detected in the one-year follow-up in ALS data compared to the baseline ALS data ($n = 11$). Color represents fiber orientations (red-left to right, blue-inferior to superior) (ALS Amyotrophic lateral sclerosis, FDC Fiber density and cross-section, FD Fiber density, FCFiber cross-section, FBA Fixel-based analysis).

posterior limb of internal capsule (bilateral findings for all the ROIs), body of corpus callosum, and the pontine crossing tract (refer Supplementary Data 1 for the full list of regions with cluster size and MNI coordinates, where decrease in FDC is detected). We did not find any longitudinal change in the control participants.

*Longitudinal data: change in FDC in slowly progressing ALS participants.* Because there is a considerable need for biomarkers of disease progression in ALS, particularly among slowly progressing clinical trial participants, we analyzed longitudinal change in FDC among the 6 study participants with a change in ALSFRS-R of < 6 points over 12 months, or an ALSFRS-R slope of < 0.5 points per month. The mean change in ALSFRS-R in this group is 1.5 with a range of 0 to 5. FDC demonstrated a statistically significant change (p-value 0.03) in this small cohort of participants with very slow clinical progression, FDC decreased in all 6 participants at 12 months.

*Longitudinal data: Multimodal analysis of brain Fsum, cortical thickness, brain FDC, spinal cord RD, and spinal cord CSA.* Longitudinal group changes in individual metrics in ALS participants and their effect sizes and p-values are shown in Fig. 6 and Table 4, respectively. We have previously reported the longitudinal changes in FA in spinal cord tracts from C2 to C6[24]. In that study, we did not identify statistically significant longitudinal change in spinal cord CSA (whole cord, white matter, or gray matter), along or averaged across the cord. Here we report the group classification performance with a multimodal analysis of 12-month changes in the brain and spine measures.

Brain fixel-based measures provided the greatest effect sizes and lowest p-values, followed by brain Fsum, in detecting

longitudinal change at 12 months, and those remained significant after Bonferroni-Holm correction. However, for group classification (ALS vs. control) based on 12-month longitudinal data, brain FA provided higher accuracy (87.5%), sensitivity (82%), and specificity (92%), compared to the accuracy, sensitivity, and specificity provided by the other measures separately and together as a multimodal analysis (Supplementary Table 1).

*Longitudinal data: Correlations with functional status.* We did not find a significant correlation between the change in any of the diffusion metrics or cortical thickness and the change in ALSFRS-R over a one-year period.

*Longitudinal data: Change in FDC at six months in ALS participants.* All the longitudinal analysis results presented above are from the baseline vs. one-year follow-up comparison. The changes in these metrics at the six-month follow-up visit were in general quite small and in some cases not linear between baseline and 12 months. However, since six month changes are of potential interest for clinical trial design[51], we performed a statistical analysis for potential changes over six months, based on the regions identified from the 12-month data. We created a mask to evaluate fixel values in voxels where statistically significant change was detected in the 12-month follow-up data, including corticospinal tract, superior corona radiata, cerebral peduncle, posterior limb of internal capsule, body of corpus callosum, and the pontine crossing tract (refer Fig. 5). This analysis detected marginally significant longitudinal changes in FC (uncorrected p-value 0.022), FD (uncorrected p-value 0.049), and FDC (uncorrected p-value of 0.045).

## Discussion

We report findings of a longitudinal multimodal MRI analysis in people with ALS and healthy control participants at enrollment, 6 months, and 12 months. We investigate diffusion and morphology measures from the brain and spine for their ability to detect disease-related changes in ALS, and we demonstrate improved sensitivity using a multimodal analysis of brain and cervical spinal cord that includes FBA metrics. Specifically, brain measures included FA, Fsum, cortical thickness, FD, FC, and FDC, and spine measures include FA and CSA of the spinal cord.

Fsum detected baseline differences between ALS and control participants in the largest number of brain regions, but FA provided higher effect size and higher statistical significance (lower p-value). This may reflect the lower variance in FA values. At the same time, Fsum resolves fiber bundles in areas with fiber crossings (e.g. centrum semiovale), which FA cannot, albeit with greater sensitivity to noise, which may be the reason for the detection of more regions. We also found that Fsum was consistent in detecting disease-related differences for both the cross-sectional group comparison and the group comparison of longitudinal changes at 12 months. The benefits of Fsum- and FA-based analyses were complementary, with Fsum being a consistent biomarker of disease progression, while FA provides a higher effect size in disease detection.

FDC performed well in detecting longitudinal change in the ALS participants. Remarkably, FDC identified statistically significant progression at 12 months in a subset of only 6 participants with unusually slow progression as measured by ALSFRS-R. Furthermore, FDC demonstrated marginally significant change at the six-month follow-up visit, further study will be required to clarify if this could be a useful biomarker during the commonly utilized 6-month timeframe of clinical trials.

Cortical thickness data demonstrated significant cross-sectional differences between ALS and control participants in the pre- and

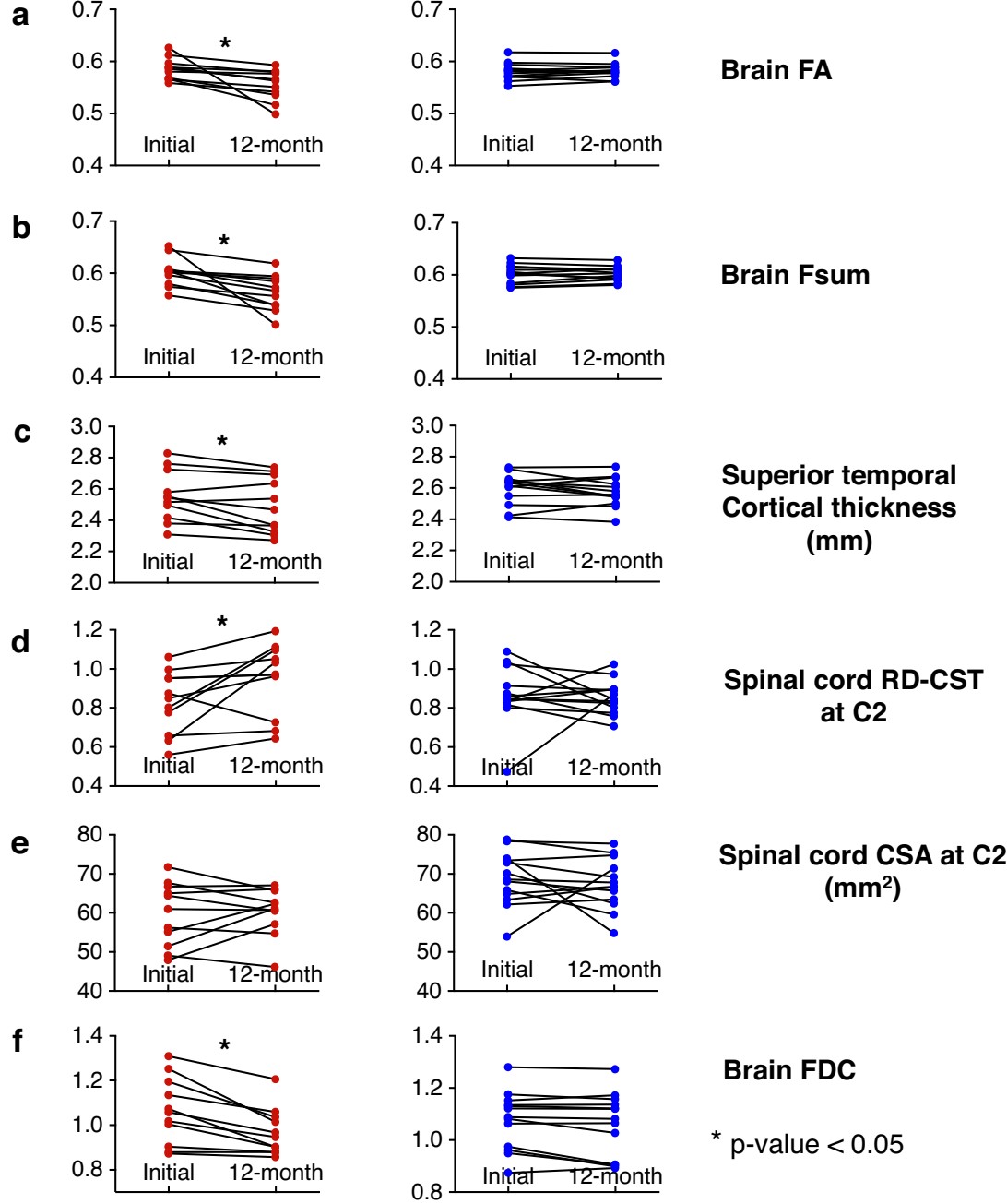

**Fig. 6 Summary of longitudinal change in different brain and spinal cord measures.** Longitudinal changes in ALS participants ($n = 11$, panels in column 1) and controls ($n = 13$, panels in column 2) in (**a**) brain FA, (**b**) brain Fsum, (**c**) superior temporal cortical thickness, (**d**) spinal cord CST RD at C2, (**e**) spinal cord CSA at C2, and (**f**) brain FDC. The corresponding effect sizes and p-values of the longitudinal change in ALS participants are provided in Table 4 (ALS Amyotrophic lateral sclerosis, FA Fractional anisotropy, Fsum Sum of fiber volume fractions, RD Radial diffusivity, CST Corticospinal tract, C2 Spinal cord C2 level, CSA Cross-sectional area, FDC Fiber density and cross-section).

post-central gyri, adjacent frontal and parietal regions, and pars opercularis. By contrast, longitudinal change in ALS was most notable in temporal regions, among others. This may be due to the fact that the earlier regions are affected in the early stages of the disease, and the later regions in later stages. Our findings are consistent with the extensive literature on cortical thickness in ALS, demonstrating frontal, temporal, and parietal involvement[52–55]. This roughly corresponds to regions identified in the pathological staging proposed by Brettschneider et al.[56]. While ALS is clinically characterized as a disorder of motor, cognitive, and behavioral function, our finding of relative atrophy in parietal regions does

correspond with clinical, physiologic, and imaging studies identifying abnormalities in sensory pathways and function[24,57–59]. We noted a large variability in individual cortical thickness in some regions, especially in frontal and temporal poles, including an increase in cortical thickness in these regions in some participants (Supplementary Fig. 4). This variability is likely due to the small size and the location of these regions which makes it challenging for the freesurfer to segment them accurately. This increase was not statistically significant, however.

We have noted trends for correlations between Fsum, FA, and ALSFRS-R in several brain regions, although those correlations

**Table 4 Effect size and p-values in ALS participants: Multimodal longitudinal 12-month change data.**

| Measure | Effect size (standardized response mean) | P-value: paired t-test for longitudinal change | Corrected p-value (Bonferroni-Holm) |
|---|---|---|---|
| Brain FA | 0.89 | 0.014* | 0.056 |
| Brain Fsum | 1.00 | 0.0078* | 0.039* |
| Superior temporal cortical thickness | 0.85 | 0.018* | 0.054 |
| Spinal cord RD CST at C2 | 0.75 | 0.033* | 0.066 |
| Spinal cord CSA CST at C2 | 0.13 | 0.680 | 0.680 |
| Brain FDC | 1.33 | 0.0013* | 0.0078* |
| Brain FC | 1.34 | 0.0012* | 0.0084* |
| Brain FD | 1.41 | 0.0008* | 0.0070* |

Mean FA and Fsum are for voxels with significant change in the TBSS analysis. See text for explanation.

*p-value < 0.05 (ALS Amyotrophic lateral sclerosis, FA Fractional anisotropy, Fsum Sum of fiber volume fractions, CST Corticospinal tract, C2 Spinal cord C2 level, CSA Cross-sectional area, RD Radial diffusivity, FDC Fiber density and cross-section).

were not significant after corrections for multiple testing. We have previously reported[24] strong correlations between spinal cord FA and CSA, and the limb components of ALSFRS-R. In the brain, we found that correlations also exist between the fixel-based metrics (FD, FC, and FDC) and ALSFRS-R (refer Fig. 3, only FDC is shown here as FD and FC are derived from FDC).

Even though we would characterize our cohort as an early-stage cohort based on the mean ALSFRS-R (40.0 at baseline), note that the disease duration varies from 3.4 to 147.7 months, with an average of 38.5 months. This shows that there are long-surviving patients in our cohort who are progressing at a rate much slower than the typical rate of progression in ALS[60]. This can have implications for the ability to detect an effect with the proposed biomarkers. This also suggests that these metrics may be highly sensitive to change in a more typical cohort, and possibly over a shorter period of observation. With regard to baseline clinical characteristics such as age (mean 57.5 years), site of onset (majority limb-onset), and gender (slight majority male), our cohort represents a typical ALS population. The sole atypical feature, slow progression, (ALSFRS-R slope of 0.40 points/month compared to the reported average change of 0.9 to 1 point/month[60]), highlights the value of our findings. Despite a small sample size of largely slow progressors, we observed a statistically significant longitudinal change in Fsum and FDC.

It is sometimes difficult to diagnose ALS in patients with a paucity of UMN signs, leading to diagnostic errors or delay. For these individuals, a paraclinical marker of UMN involvement is needed. Notably, we also found a significant difference between 13 ALS participants with low UMN scores, indicating limited clinical evidence of UMN disease, and the control cohort. This merits further refining for comparison between ALS patients and patients with other neurologic diseases that can mimic ALS as controls, to determine if it might have value as a diagnostic marker at the single-patient level.

Notably, brain FA, Fsum, and cortical thickness were lower in the 6 LMN-predominant ALS participants than in controls, and this difference was highly significant. The reason for this is unclear. While one could speculate that it supports the view that UMN pathology is the principal driver of disease in ALS, it may simply reflect the fact that severe weakness is a marker of more advanced disease, and that UMN signs are masked by weakness at this point.

The multimodal analysis with brain FA, cortical thickness, spinal cord FA, and spinal cord CSA provided the best group classification accuracy, sensitivity, and specificity in the cross-sectional data. We report 90% sensitivity and 85% specificity using the multimodal metrics from dMRI and T1 of the brain and the dMRI of the spinal cord. This is similar to values reported by

Querin et al.[26] utilizing dMRI, T2, and MTR of the spinal cord, and Bede et al.[61], using dMRI and T1 of the brain. They used predictor variable selection based on correlations and discriminatory significance. Using a logistic regression model for classification, we also demonstrate the added benefit of a multimodal analysis and compare the diagnostic performance of the metrics we used.

In another multimodal study of DTI, MTR, and R2* Borsodi et al.[27] concluded that DTI better detected white matter changes compared to MTR and R2*. They found no correlation between tract-specific diffusion properties at baseline and the clinical status measured with ALSFRS-R. In our study, we also noted that diffusion metrics are better at detecting disease-related changes in ALS compared to cortical thickness and spinal cord CSA and, using fixel-based methods, found a correlation between FDC and the ALSFRS-R.

In their multimodal study, van der Burgh et al.[25] reported that spinal onset participants had more white matter involvement and bulbar onset participants had more gray matter involvement at baseline. Here, we identified more white and gray matter involvement in the spinal onset participants, with more gray matter involvement in lower limb onset participants compared to upper limb onset participants.

Single imaging measures are found to be more effective than multimodal markers in detecting disease-related longitudinal changes with the metrics and analysis methods we used in this study. In particular, we found fixel-based metrics superior at detecting longitudinal change. Further studies are needed to confirm this finding, or whether a multimodal approach with additional metrics might help further improve detection of longitudinal changes.

In both the cross-sectional and longitudinal analyses, we first performed whole brain analysis to identify brain regions with statistically significant group differences (or change, in the longitudinal case). Following this step, we extracted the mean values of imaging metrics (e.g., FA, Fsum, and FDC) from those regions, for use in the multimodal analysis. Our multimodal analysis results are not particularly sensitive to the mask we used to extract the mean values, the mask actually covers the affected regions we listed in the results section, one can reproduce our results simply by using an ROI mask corresponding to those regions we listed (e.g., corticospinal tract, corpus callosum).

Our study, like many others, is limited by the relatively small number of participants and high drop-out over the course of 12 months. This remains a shortcoming of ALS imaging studies with relatively long periods of follow-up. Also, the site of onset analysis and other subgroup analyses we conducted have more of an exploratory nature due to the limited number of participants

in each subgroup. Nonetheless, given the consistency of our results across imaging modalities and analysis methods, as well as our thorough statistical analysis, we believe that our findings are reliable and useful to inform us about the pathophysiology of ALS.

The need for biomarkers of disease progression is well-recognized in ALS clinical research. The universally used functional outcome measure, ALSFRS-R, has several shortcomings that are well-documented in reviews of ALS research[5–7]. Fluid biomarkers, most notably neurofilament light (NfL), have received considerable interest but have not yet shown sufficient sensitivity to change over the time course of most clinical trials. At the same time, current FDA policies encourage the development of biomarkers to accelerate drug development[62]. Therefore, a multimodal imaging metric that could demonstrate sensitivity to change over short durations with a large effect size would be highly impactful in ALS research. Insofar as our protocol combines data from the brain and spinal cord C2 level, it has the potential to provide multimodal data from both brain and spinal cord in a single, scalable imaging study. Note that we used only the spinal cord data from the C2 level in this work, whereas we reported the data from C2 to C6 in our prior work[24]. We noted similar diagnostic performance with mean C2-C6 data (compared to C2 data only). We therefore chose to include only the C2 data in this work to demonstrate the potential to conduct future multimodal studies using a single head and neck scan.

In summary, neuroimaging data has previously only demonstrated limited utility as a diagnostic biomarker in ALS, or as a biomarker of disease progression. By analyzing dMRI metrics capable of resolving crossing fibers, tract-based methods, and morphometry, as well as incorporating data from both brain and cervical cord, we demonstrate change over 12 months in a small cohort of participants with slowly progressive ALS. Our approach also validates the high sensitivity and specificity of multimodal imaging in distinguishing participants with ALS from controls and, notably, distinguishes a cohort of only nine participants with minimal UMN signs from controls.

Future work will aim at refining the imaging protocol, that is both comprehensive enough to leverage the modalities we studied and clinically feasible. Prospective application of such a protocol is needed to confirm its utility as an adjunct in the diagnostic process and a biomarker of ALS disease progression.

## Data availability

Deidentified data will be made available on request for the purposes of reproducing the results presented, subject to institutional approval. The source data underlying Figs. 2, 3, and 6 are made available as Supplementary Data 2.

## Code availability

The code used to produce the results reported in this article is publicly available in ref. [63].

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

## Acknowledgements

P.K.P. is funded in part by grant number 2020-225709 from the Chan Zuckerberg Initiative DAF, an advised fund of Silicon Valley Community Foundation. This study was also supported by funding from Bob Allison Ataxia Research Center (BAARC), the Curt O'Hagan ALS/PLS and ALS-Lou Gehrig funds of the University of Minnesota Foundation, NIH grants P41 EB015894 and P30 NS5076408. We are grateful to the persons living with ALS, their families, and the control participants who volunteered their time; without them this work would not be possible. We acknowledge Valerie Ferment and Susan Rolandelli, RN, for study coordination, and Pamela Droberg, NP, for her assistance in recruitment.

## Author contributions

P.K.P., L.E.E., D.W., and C.L. contributed to the study concept and design; P.K.P., L.E.E., I.M.A., G.M., G.G., D.W., and C.L. contributed to the data acquisition and analysis; P.K.P., L.E.E., I.M.A., D.W., and C.L. contributed to drafting a significant portion of the manuscript and figures, and all authors contributed to revision of the paper and to the final draft. D.W. and C.L. are joint senior authors.

## Competing interests

The authors declare no competing interests.
