## [Peer Review File · Communications Medicine]

Reviewers' comments:

Reviewer #1 (Remarks to the Author):

This work reports results from a study analyzing cross-sectional DTI, structural metrics (cortical thickness) and fixed-based microstructure metrics of the brain and cervical spine of 20 ALS patients and 20 controls. An additional longitudinal analysis of 10/20 ALS and 14/20 controls at 1-year is also described. By using these metrics, an increased diagnostic accuracy in differentiating ALS from controls is reached respect to a previous work from the same group in which only diffusion and cross-sectional area of the cord were used. In this work, fiber density and cross-section provided the greatest sensitivity in differentiating HC from ALS and evidence of progression was demonstrated both with spinal and cerebral measures.

The paper has different limitations, a few queries are listed below:

- Part of the results, sometimes in the form of entire paragraphs have already been published by the same group (Pishsarady et al, Commun Biol 2020) and inserted in this manuscript.
- Sample size: sample size of only 20 ALS patients (of which only 14 has a second scan) is quite scarce. Furthermore, when dividing patients according to site of onset (spinal vs bulbar) the significance of the results loses power;
- Cross-sectional data: spinal cord FA and CSA at C2: authors refer to results published in a previous study, in which they analyzed MRI metrics of the entire cervical spine, stating that in this current work only cross-sectional area of the CST at the C2 level will be reported. However, the choice of reporting only data regarding the C2 level should be explained. Figure 2 (D, E) also reports the differences in MRI metrics only at the C2 level. Does that mean that other levels produced non-significant differences? The decision to present only these results should be explained.
- Cross sectional data: distinguishing participants with UMN burden from controls. The difference in MRI metrics between ALS patients with UMN burden score of 1 or 2 respect to HC is reported. However, it would be interesting to analyze also differences in ALS patients with low and high UMN burden, which is lacking.
- Table 2: brain FA and brain Fsum. In the table it is reported that Brain FA and Fsum are calculated as the mean of the statistically significant voxels in the TBSS analysis, However, this should be made explicit also in the main text of the manuscript, in the methods section.
- Longitudinal data: fiber density and fiber cross-section: the significant longitudinal differences in fiber density and fiber cross-sections of the tracts in the ALS cohort is reported, however p values are missing.
- Longitudinal data: change in FDC in slowly progressing ALS patients. What emerges from this paragraph is that at least one patient had a higher ASL-FRSr at the 1-year follow-up visit of at least 2 points (mean

change in the group of 1.5, range of -2 to 5). The single case should be discussed in the appropriate section and an explanation for this improvement in functionality should be provided.

- Genetic analysis: no statement about the genetic status of the patients is reported.
- Table 1: quality of the table could be improved (e.g. no background, neutral colors)

Reviewer #2 (Remarks to the Author):

- Please add in the keywords list: longitudinal study, disease progression.

Introduction

- "Quantification of the changes in diffusion using dMRI is therefore a surrogate marker of axonal degeneration" diffusion MRI is not a specific to axonal degeneration (see Concha L. Neuroscience 2014).

- No need for general concepts related to diffusion. Please focus on ALS and the contribution of MRI (dMRI) in explaining the pathophysiology of the disease then state your work hypothesis.

- Your introduction is brain focussed and should take into account spine studies in which DTI was successfully applied in ALS despite the methodological limitations. Besides, your study is omitting citing the most important studies in ALS investigating spinal cord cross-sectional/longitudinal damage and its impact on patient's functional scores and worsening (see the review from El Mendili MM et al., 2019).

- Please make your introduction shorter and introduce only the key points that justify the need of another investigation of the brain/spine in ALS using MRI.

Methods

- Please give a brief description of what was previously done at the spine level (Pisharady PK et al., 2020).

- Were p-values from the single modality analysis corrected for multiple-comparisons (TBSS, results illustrated in Figure 1) ?

- “The cortical thickness measure used for cross-sectional multimodal analysis is the cortical thickness of the precentral area and that for longitudinal multimodal analysis is the cortical thickness of the superior temporal area (average of left and right ROIs in both cases).”

Could you please justify the methodological choice/difference.

- In your previous article, you analysed a large portion of the cervical spinal cord (C2-C6), while here you are only quantifying CSA/FA at C2 vertebral level.

Could you please justify this methodological choice.

- Table 1, please provide the clinical and demographical data for the 6mo follow-up visit.

- In “The multimodal analysis was done using multivariable logistic regression with the disease status as the response variable”, what do you mean by disease status?

Please be more specific in the main text.

- Correlations: Could you please indicate the method that was used for correlations. In addition, could you please perform the correlations for the 6mo follow-up.

Results

- Please provide supplementary tables for TBSS, cortical thickness and FBA results with the following elements:

+ Regions with significant clusters (using an appropriate atlas)

+ Cluster size

+ Minimal p-value

+ MNI coordinates

- Please replace “@” by “at” in Figure 2 and define ALS/C2/Fsum in the figure’s legend.

- Please define all used abbreviations in the legend of each figure/table.
- Supplementary figure 2 shows important results. I would advice to put it in the main document.
- “Longitudinal data: FA and CSA along the spinal cord” sub-section: the results of your previous study should not be placed in the results section but need to be discussed.

Discussion

- Due to your sample size, you need to be more cautious and weight up your arguments when discussing results from a sub-group of your ALS population.

Reviewer #3 (Remarks to the Author):

Summary of the study

In this study, the authors used a multimodal approach to identify the cortical and spinal cord measures that are altered in ALS. Clinical data (ALSFRS-R, ECAS) were collected, and multimodal MRI was done at baseline and for a subset of participants at longitudinal visits at month 6 and month 12. Neurological examinations were conducted at baseline and 12 months. The study was performed on two scanners: Siemens Trio and Prisma. The authors used brain diffusion and cortical thickness measures to identify alterations in cortical gray and white matter. Additionally, the authors present a relatively novel method of assessment of brain white matter microstructure in ALS using a fixel-based approach. An assessment of MRI measures and their correlations with clinical variables (ALSFRS-R) were performed at baseline and at longitudinal follow up time points of 6 and 12 months.

The main findings were that DTI- and fixel-based measures indicated alterations in white matter microstructure (reduced FA and Fsum) in motor and extra-motor regions in ALS at baseline and longitudinally. There was cortical thinning at baseline in the precentral gyrus. Longitudinal changes in cortical thickness did not survive corrections for multiple comparisons. No significant alterations of brain FDC (combined fiber density and cross-section) were observed at baseline. However, FDC declined

longitudinally in white matter tracts subserving the cortical motor regions. Specifically, FDC was declined in slowly progressing patients (functional decline < 0.5 points/month). Significant clinical correlations were found between ALSFRS-R and mean FDC of the right and left corticospinal tract at baseline. No significant correlations were observed between imaging and clinical measures longitudinally. At the 6-month follow up timepoint, marginally significant alterations in fiber density and fiber cross-section values were observed. Predictive analyses were performed at baseline and follow up. These models utilized individual imaging metrics, either individually or as a combination, to predict disease status.

In summary, the authors conclude that, in a slowly progressing ALS cohort, a prediction model with multiple MRI modalities provided the best performance in diagnostic classification cross-sectionally, and that fixel-based measures had the potential to identify longitudinal changes in white matter microstructure.

This manuscript had a number of strengths:

1. The methods were well described and clear.
2. There was a multiregional approach, with MRI of both the brain and spinal cord. The study is relatively unique in this respect.
3. The authors present raw data points, for example, for cross sectional and longitudinal data, as well as for individual subject cortical thickness maps.
4. Multiple modalities were used to assess brain microstructure and cortical thickness to interrogate different aspects of neurodegeneration.

The reviewers propose some major and minor considerations that would be helpful in improving the quality of the manuscript.

Major revisions

1. The reviewer would have preferred to see the objectives and hypotheses stated at the end of the Introduction rather than a summary of the methods and findings of the study. If there were no a priori objectives/hypotheses, it would be important to clarify that in the introduction.
2. It is notable that there is a very large range in disease duration with the inclusion of essentially “long survivors”. Additionally, the disease duration of patients is greater than 3 years on average which is not considered early stage.
 - a. Thus, there are patients that are progressing much slower than the average person with ALS which can have implications for the ability to detect an effect with these proposed biomarkers; there may have

been a longitudinal change observed in the MRI biomarkers ALS patients with a typical rate of functional decline. This should be noted in the Discussion.

b. Are the authors suggesting that within the slowly progressing ALS cohort, there are also early and late stages?

3. The ALSFRS-R upper limb “sub-score” is incorrect; the range should be 0-12 not 0-36.

4. Page 7, lines 10-11: A suggestion for the authors would be to summarize the findings on spinal cord cross-sectional FA and CSA from their previous publication. This would be helpful to interpret the findings from the current study.

5. Page 7, lines 22-23: Low UMN burden scores do not automatically indicate lower motor neuron predominance. For patients with low UMN burden scores, it is also suggested that the authors present LMN-specific data. Is there a LMN burden score? Patients with high LMN burden scores and low UMN burden scores would be considered to have LMN-predominant ALS.

6. It is unusual to see an increase in ALSFRS-R after 12 months. This is a reflection of the slowly progressing and unusual ALS cohort in this study which is probably very heterogeneous. Further, for this single individual, could the ALSFRS-R have increased due to symptomatic therapy (for example, of sialorrhea or spasticity)?

7. The sample size in this study is very small. It is suggested that the authors discuss the implications of that in more detail in the discussion in terms of the reliability of identifying biological mechanisms of disease and limitations arising due to small sample sizes.

8. The authors mention that post hoc analysis of scanner effects across visits did not have a significant effect on the data reported in the study. A description of the analyses performed, and the outcomes should be added to the study.

9. It is the preference of the reviewers that discussion points should not be included in the results. Therefore, it is suggested that the authors' qualification of the Fsum findings in the results should be moved to the Discussion section.

10. The 95% confidence intervals appear to be inaccurate in figure 2. It is recommended that the authors re-generate these plots with accurate estimates of confidence intervals.

11. Clinical correlations: As the ALSFRS-R score is largely driven by the impairment of lower motor neurons, it is suggested that the authors report correlations between brain metrics and clinical measures pertaining to upper motor neurons, for example, with UMN burden scores.

12. It is commendable that the authors present individual subject data on longitudinal cortical thickness measurements in the supplementary material. Although cortical thickness measurements show a marginally significant reduction over a 1 year follow up in ALS, the figures in the supplementary section indicate that there is cortical thickening over time. We suggest that the authors clarify the discordance between actual cortical thickness alterations and their interpretation of these alterations. It is also suggested that the authors consider presenting increases/decreases in cortical thickness as separate figures.

Minor revisions

1. The authors mention the use of a ball-and-stick model (TBSS) as compared to the tensor model (DTI) (page 3, line 19). It would be helpful for the authors to clarify the rationale for the choice of a TBSS over DTI model. A short description of the novel fixel-based measures and their relevance to health and disease would be useful.
2. It is very interesting to note that none of the patients in King's stage 3/4 at baseline returned for a follow up scan at 12 months. It would be important to see how the patients at King's stage 1/2 are different from patients at stage 3/4, in terms of MRI metrics and demographic characteristics at baseline. However, it is accepted that this would be an exploratory analysis given the small sample size.
3. In addition to the exploratory analyses presented of spinal vs bulbar ALS, it would be interesting to explore disease heterogeneity with a comparison of "fast" and "slow" progressors.
4. Was a cutoff employed for the ECAS total score to select healthy participants for inclusion in the study?

Improved Diagnostic Accuracy and Sensitivity to Longitudinal Change in ALS with Multimodal MRI of the Brain and Cervical Cord

Response to review comments

We thank all the three Reviewers for their valuable and insightful comments, which helped us improve the manuscript. We have completed a substantial revision of the manuscript, and added content to the results, discussion, methods, and introduction sections, based on the comments we received (major additions are highlighted in blue in the manuscript). We addressed all the concerns the reviewers expressed, as detailed in this document. The major changes we made include additional analysis based on LMN score, additional analysis based on King's staging, inclusion of 6-month clinical and demographic data, addition of several discussion points as requested, revision of the introduction section, addition on the description of fixel-based analysis, summarizing our previous work, and the addition of supplementary tables presenting the cluster size and coordinates of the regions affected. Below we provide point-by-point answers to the comments we received.

Reviewer #1

General comment: *This work reports results from a study analyzing cross-sectional DTI, structural metrics (cortical thickness) and fixed-based microstructure metrics of the brain and cervical spine of 20 ALS patients and 20 controls. An additional longitudinal analysis of 10/20 ALS and 14/20 controls at 1-year is also described. By using these metrics, an increased diagnostic accuracy in differentiating ALS from controls is reached respect to a previous work from the same group in which only diffusion and cross-sectional area of the cord were used. In this work, fiber density and cross-section provided the greatest sensitivity in differentiating HC from ALS and evidence of progression was demonstrated both with spinal and cerebral measures.*

The paper has different limitations, a few queries are listed below:

Our response: We thank the Reviewer for the comments and for pointing out the limitations. Please note that the increased diagnostic accuracy we reported is not in comparison with our previous work that used only spinal cord data. We are reporting the disease diagnosis performance specific only to this work; the improvement we are reporting is in the multimodal analysis compared to single modality analysis (e.g. brain FA + cortical thickness vs. brain FA alone).

Comment-1: *- Part of the results, sometimes in the form of entire paragraphs have already been published by the same group (Pishsarady et al, Commun Biol 2020) and inserted in this manuscript.*

Our response: The part the Reviewer is referring to here is the subsection (two paragraphs, and Table 1) that report the cohort characteristics. The cohort and cohort characteristics in this work and our previous work remain the same, but the data and results we present are entirely different. Nonetheless, to answer the

Reviewer's concern, we made the following changes: First, we have added 6-month visit summaries to the table (as requested by Reviewer 2). Second, our prior paper (Pisharady et al, Commun Biol. 2020) presented a *tract-specific* and *along-the-tract* (C2 to C6) analysis of spinal cord data, whereas this paper presents a multimodal analysis of brain and spine data (C2, also please refer to our response to Comment 3). We have emphasized this in the manuscript. Finally, we acknowledge the use of the same cohort and repetition of the Table in the paper as follows:

“We have reported the demographic and clinical features of this cohort in our prior publication which focused on spinal cord data²⁴. The table presenting demographic and clinical features, updated with 6-month data, is included here (Table 1)”

We have also taken care to avoid verbatim, but we repeated the subtitle “Cohort characteristics” for this subsection (which caused the above confusion). We have revised this subsection carefully to minimize repetitions (except the subtitle and parts of Table 1).

Comment-2: - *Sample size: sample size of only 20 ALS patients (of which only 14 has a second scan) is quite scarce. Furthermore, when dividing patients according to site of onset (spinal vs bulbar) the significance of the results loses power;*

Our response: We had 20 ALS and 20 control subjects at baseline, and 11 ALS and 13 control subjects at one-year follow-up. Despite the relatively small sample size, we believe that factors such as the quality of the data (high spatial and angular resolution, collected using advanced protocols and corrected for distortions using state-of-the-art techniques), and the robustness of the methods we used led to important findings. We agree that the sample size, especially in the longitudinal analysis, is small. We also acknowledge that drop-out due to disease progression is an inherent problem in longitudinal studies in ALS. We have described this as a limitation of our study in the ‘Discussion’ section:

“Our study, like many others, is limited by the relatively small number of participants and high drop-out over the course of 12 months. This remains a shortcoming of ALS imaging studies with relatively long periods of follow-up.”

For the site of onset analysis, we had 15 spinal onset and 5 bulbar onset participants; we agree that these numbers are low, however, we think it is still worth reporting these results and have added the following sentence to the above paragraph.

“Also, the site of onset analysis and other subgroup analyses we conducted have more of an exploratory nature due to the limited number of participants in each subgroup.”

As an exploratory analysis, we feel this serves as an important proof of concept for subsequent, larger studies. In addition, the findings in our small sample suggest a large effect size, providing motivation to pursue this question on site of onset in further research.

Comment-3: - *Cross-sectional data: spinal cord FA and CSA at C2: authors refer to results published in a previous study, in with they analyzed MRI metrics of the entire cervical spine, stating that in this current work only cross-sectional area of the CST at the C2 level will be reported. However, the choice of reporting only data regarding the C2 level should be explained. Figure 2 (D, E) also reports the differences in MRI metrics only at the C2 level. Does that mean that other levels produced non-significant differences? The decision to present only these results should be explained.*

Our response: The Reviewer is right that we have used only the data from the C2 level in this study, whereas our prior paper reported data along the tract, from C2 to C6. We did so due to two reasons:

1. In this study, we want to demonstrate the potential of multimodal data from both the brain and spinal cord, that could be acquired in a single, scalable imaging study using a combined head and neck coil. This is now mentioned in the Discussion.
2. In our prior paper (Pisharady et al, Commun Biol. 2020), we already reported the detailed analysis of C2-C6 spine data. We noted that some of the metrics (e.g. whole cord FA, that is without tract segmentation) are more sensitive at C2 compared to other levels. However, the measures we used in this work (e.g. FA in CST at C2 and CSA at C2) are equally sensitive at other levels. We have conducted an exploratory analysis with these data from C2 to C6, and obtained very similar diagnostic performance. As such, we have decided to include only C2 data in this paper. We have added the below text to the Discussion section to clarify this.

“Note that we used only the spinal cord data from the C2 level in this work, whereas we reported the data from C2 to C6 in our prior work²⁴. We noted similar diagnostic performance with mean C2-C6 data (compared to C2 data only). We therefore chose to include only the C2 data in this work to demonstrate the potential to conduct future multimodal studies using a single head and neck scan.”

Comment-4: - *Cross sectional data: distinguishing participants with UMN burden from controls. The difference in MRI metrics between ALS patients with UMN burden score of 1 or 2 respect to HC is reported. However, it would be interesting to analyze also differences in ALS patients with low and high UMN burden, which is lacking.*

Our response: We thank the Reviewer for this suggestion and have conducted this additional analysis, comparing the participants with low (1 or 2) and high (3-6) UMN burden scores. However, we did not find

statistically significant differences in any of the compared metrics. We have added the below sentence to the section the Reviewer referred to.

“We did not find statistically significant differences in any of these metrics between low and high UMN burden score participants.”

Comment-5: - *Table 2: brain FA and brain Fsum. In the table it is reported that Brain FA and Fsum are calculated as the mean of the statistically significant voxels in the TBSS analysis, However, this should be made explicit also in the main text of the manuscript, in the methods section.*

Our response: We have stated this explicitly both in the Results section (where Table 2 is referred to) and in the Methods section (in the subsection ‘Statistical analysis of multi-modal data’).

Results section: “The metrics used are mean FA extracted from the voxels from TBSS analysis where we found statistically significant group differences between ALS and control participants (which cover affected regions listed in the TBSS results section, including corticospinal tract and corpus callosum), cortical thickness of the precentral area, FA in CST at C2 level, and CSA at C2 level. We first present the unimodal analysis results with these metrics as well as with Fsum (which is extracted in a similar manner as FA).”

Methods section: “ In order to extract brain and spine metrics for the multimodal analysis, we used the following strategy. For brain FA and Fsum, we created an ROI mask of the voxels with statistically significant group difference between ALS and control participants (in the case of cross-sectional analysis) or between time points for ALS participants (in the case of longitudinal analysis), as detected by the TBSS analysis. Mean values for the metrics of interest were extracted, within this mask, and were used for subsequent analyses.”

Comment-6: - *Longitudinal data: fiber density and fiber cross-section: the significant longitudinal differences in fiber density and fiber cross-sections of the tracts in the ALS cohort is reported, however p values are missing.*

Our response: We have now reported the p-values and effect sizes for fiber density (FD) and fiber cross-sections (FC) which are quite comparable to that for FDC. We had not reported those originally as both FD and FC are already represented by the fiber density and cross-section (FDC). Including the correlated measures of FC and FD separately also increases the number of variables tested, making correction for type I error for multiple testing even more challenging. We have now added those two variables to our analyses, included them in the type I error correction, and reported these results in Table 4.

Comment-7: - *Longitudinal data: change in FDC in slowly progressing ALS patients. What emerges from this paragraph is that at least one patient had a higher ASL-FRSr at the 1-year follow-up visit of at least 2*

points (mean change in the group of 1.5, range of -2 to 5). The single case should be discussed in the appropriate section and an explanation for this improvement in functionality should be provided.

Our response: We thank the Reviewer for asking about this particular participant. This participant had the same ALSFRS-R at 12 months (a score of 45) as at baseline. As such the change in ALSFRS-R for this subject at 12-months is actually zero. At 6 months, this participant had an ALSFRS-R of 47 which was mistakenly taken as 12-month data. From baseline to 6 months, there was therefore an increase of 2 in ALSFRS-R, which likely reflects test-retest variability. We corrected this error in the paper, and also double checked the ALSFRS-R data for the other participants.

Comment-8: - Genetic analysis: no statement about the genetic status of the patients is reported.

Our response: Thanks to the Reviewer for pointing this out. We have added the following sentence to the paper:

“No ALS participants had a documented history of ALS in first-degree relatives. Three had negative genetic testing using an ALS gene panel, two tested negative for C9orf72 repeat expansions. The others did not undergo genetic testing.”

Comment-9: - Table 1: quality of the table could be improved (e.g. no background, neutral colors)

Our response: We have improved Table 1, and removed the background shade of all the Tables including Table 1.

Reviewer #2

Comment-1: - Please add in the keywords list: longitudinal study, disease progression.

Our response: Thank you for the suggestion, we have added these keywords.

Introduction

Comment-2: - “Quantification of the changes in diffusion using dMRI is therefore a surrogate marker of axonal degeneration” diffusion MRI is not a specific to axonal degeneration (see Concha L. Neuroscience 2014).

Our response: We thank the Reviewer for providing this additional reference. We have updated the sentence as shown below and also included this new reference.

“Quantification of the changes in diffusion using dMRI represents the changes in anisotropic characteristics due to degeneration of axonal membranes, myelin sheaths, or the changes in axon density and coherence²”

Comment-3: - No need for general concepts related to diffusion. Please focus on ALS and the contribution of MRI (dMRI) in explaining the pathophysiology of the disease then state your work hypothesis.

Our response: We have removed the sentences that state general concepts related to diffusion (except for the first 3 sentences including the above sentence) from the introduction. The introduction has been thoroughly revised focusing on the general contribution of brain and spine dMRI in explaining and following the pathophysiology of ALS. We also stated our hypothesis as below:

“We hypothesize that combining diffusion and morphometry measures from the brain and spinal cord in a multimodal analysis will improve the discrimination power between ALS and control participants.”

Comment-4: - Your introduction is brain focussed and should take into account spine studies in which DTI was successfully applied in ALS despite the methodological limitations. Besides, your study is omitting citing the most important studies in ALS investigating spinal cord cross-sectional/longitudinal damage and its impact on patient’s functional scores and worsening (see the review from El Mendili MM et al., 2019).

Our response: Thanks to the Reviewer for suggesting this important reference that we missed (we have cited other major works from this group); We added this reference and elaborated on the spine studies. In general, we have limited the discussion on spinal cord studies as we have a previous paper on this that we cited here. Our focus here is on discussing previous multimodal studies. However, we added the below paragraph as per the suggestion from the Reviewer.

“Mendili et al. provided a comprehensive review of spinal cord imaging methods, their advantages, and drawbacks¹⁷. They discussed magnetization transfer imaging, MR spectroscopy, and functional MRI in addition to dMRI methods. We have provided a detailed overview of spinal cord dMRI studies¹⁸⁻²³ in ALS in our prior work²⁴ [Pisharady et al., 2020]. In our spinal cord MRI paper, we reported the results of a tract-specific and along-the-tract analysis of different spinal cord measures including FA and CSA, from C2 to C6. We demonstrated that the tract-specific analysis with segmentation of ascending and descending tracts in the spinal cord white matter substantially increases the sensitivity of dMRI to disease-related changes in ALS. We identified the tracts and spinal levels affected in ALS, and reported the involvement of sensory pathways in ALS. We also noted strong correlations between spinal diffusion and cross-sectional area metrics and the spinal components of ALSFRS-R. In this study, our focus is on multimodal analysis of brain and spinal cord measures.”

Comment-5: - Please make your introduction shorter and introduce only the key points that justify the need of another investigation of the brain/spine in ALS using MRI.

Our response: We have carefully revised the introduction section and made it shorter, as per the Reviewer's suggestion. We have removed the mentions of general concepts and issues related to dMRI and avoided the description of our work (provided only the hypotheses/ motivations) in the introduction. [Please also refer to our responses to the above 2 comments].

Methods

Comment-6: - Please give a brief description of what was previously done at the spine level (Pisharady PK et al., 2020).

Our response: We have added the below description of our prior work in the Introduction section.

“In our spinal cord MRI paper, we reported the results of a tract-specific and along-the-tract analysis of different spinal cord measures including FA and CSA, from C2 to C6. We demonstrated that the tract-specific analysis with segmentation of ascending and descending tracts in the spinal cord white matter substantially increases the sensitivity of dMRI to disease-related changes in ALS. We identified the tracts and spinal levels affected in ALS, and reported the involvement of sensory pathways in ALS. We also noted strong correlations between spinal diffusion and cross-sectional area metrics and the spinal components of ALSFRS-R.”

Comment-7: - Were p-values from the single modality analysis corrected for multiple-comparisons (TBSS, results illustrated in Figure 1)?

Our response: Yes, the TBSS results presented in Figure 1, as well as the single modality results presented in Table 2, are corrected for multiple comparisons. We have added the below sentence to the Figure 1 and Figure 4 captions to clarify this.

“Color-coded regions show the areas where statistically significant group differences exist ($p < 0.05$, corrected for multiple comparisons across space).”

Comment-8: - “The cortical thickness measure used for cross-sectional multimodal analysis is the cortical thickness of the precentral area and that for longitudinal multimodal analysis is the cortical thickness of the superior temporal area (average of left and right ROIs in both cases).”

Could you please justify the methodological choice/difference?

Our response: Our objective in this study is to demonstrate the improvement achievable with a multimodal analysis. For this, we have selected the measures for multimodal analysis based on our findings from the single modality analysis. In the case of cortical thickness, the precentral cortex is the area where we found that cortical atrophy occurred the most in the baseline cross-sectional data. However, we did not find cortical thinning in the precentral cortex longitudinally. We chose the superior temporal cortex in the longitudinal case, as we found the most change in thickness in this area (only marginally significant). This may be due to

the fact that the precentral cortex is affected in the early stage of the disease, and the superior temporal cortex in later stages. We have added the below sentence to the Methods section (next to the sentence referred to here) to clarify our selection of areas.

“We chose these areas for the multimodal analysis as we found cortical thinning in these areas to be most significant in the corresponding cross-sectional and longitudinal single modal analyses.”

We also added the below sentences to the Discussion section.

“Cortical thickness data demonstrated significant cross-sectional differences between ALS and control participants in the pre- and post-central gyri, adjacent frontal and parietal regions, and pars opercularis. By contrast, longitudinal change in ALS was most notable in temporal regions, among others. This may be due to the fact that the earlier regions are affected in the early stages of the disease, and the later regions in later stages.”

Comment-9: - In your previous article, you analysed a large portion of the cervical spinal cord (C2-C6), while here you are only quantifying CSA/FA at C2 vertebral level. Could you please justify this methodological choice.

Our response: Please refer to our response to Reviewer 1, comment 3 (copied below).

The Reviewer is right that we have used only the data from the C2 level in this study, whereas our prior paper reported data along the tract, from C2 to C6. We did so due to two reasons:

1. In this study, we want to demonstrate the potential of multimodal data from both the brain and spinal cord, that could be acquired in a single, scalable imaging study using a combined head and neck coil. This is now mentioned in the Discussion.
2. In our prior paper (Pisharady et al, Commun Biol. 2020), we already reported the detailed analysis of C2-C6 spine data. We noted that some of the metrics (e.g. whole cord FA, that is without tract segmentation) are more sensitive at C2 compared to other levels. However, the measures we used in this work (e.g. FA in CST at C2 and CSA at C2) are equally sensitive at other levels. We have conducted an exploratory analysis with these data from C2 to C6, and obtained very similar diagnostic performance. As such, we have decided to include only C2 data in this paper. We have added the below text to the Discussion section to clarify this.

“Note that we used only the spinal cord data from the C2 level in this work, whereas we reported the data from C2 to C6 in our prior work²⁴. We noted similar diagnostic performance with mean C2-C6 data (compared to C2 data only). We therefore chose to include only the C2 data in this work to demonstrate the potential to conduct future multimodal studies using a single head and neck scan.”

Comment-10: - Table 1, please provide the clinical and demographical data for the 6mo follow-up visit.

Our response: We have added the 6-month clinical and demographic data to Table 1.

Comment-11: - In “The multimodal analysis was done using multivariable logistic regression with the disease status as the response variable”, what do you mean by disease status? Please be more specific in the main text.

Our response: The disease status we are referring to here is the binary status of the participants, whether patient or control. We have clarified this as below.

“The multimodal analysis was done using multivariable logistic regression with the disease status (patient or control) as the response variable.”

Comment-12: - Correlations: Could you please indicate the method that was used for correlations. In addition, could you please perform the correlations for the 6mo follow-up.

Our response: The correlation is estimated by calculating the Pearson correlation coefficient between ALSFRS-R and mean FDC in left and right CST at baseline. We found significant correlations in the baseline data that we reported. However, we did not find correlations between change in ALSFRS-R and change in MRI measures at 6 months or 12 months (as we already reported). Based on this comment, we checked the correlations between the ALSFRS-R at 6 months and the MRI measures at 6 months, and did not find statistically significant correlations. We also did not find correlations between ALSFRS-R at 12 months and MRI measures at 12 months (note that we have only 10 subjects at 6 months and 11 subjects at 12 months, whereas we have 20 subjects at baseline).

We have added the below sentence to the ‘Correlations with functional status’ subsection in Methods section.

“The correlation is estimated by calculating the Pearson correlation coefficient between ALSFRS-R and mean FDC in left and right CST.”

Results

Comment-13: - Please provide supplementary tables for TBSS, cortical thickness and FBA results with the following elements:

- + Regions with significant clusters (using an appropriate atlas)
- + Cluster size
- + Minimal p-value
- + MNI coordinates

Our response: We have added three tables (Supplementary table 1-3) to the supplemental material with the requested details for cross-sectional and longitudinal TBSS analysis, and the longitudinal FBA analysis. We have referenced these Tables in the results section of the manuscript and described how the cluster size calculations are done in the methods section. This comment is not applicable for the cortical thickness analysis as the analysis is done using the mean thickness of the cortical regions and as there is only one p-value corresponding to each region (which is already reported in the manuscript).

Comment-14: - Please replace “@” by “at” in Figure 2 and define ALS/C2/Fsum in the figure’s legend.

Our response: We have made this change in Figure 2 (and Figure 6) and defined all the abbreviations.

Comment-15: - Please define all used abbreviations in the legend of each figure/table.

Our response: We have defined all the abbreviations in the caption of each figure/table.

Comment-16: - Supplementary figure 2 shows important results. I would advice to put it in the main document.

Our response: Supplementary figure 2 shows individual participant data of whole brain cortical thickness change, whereas the reported statistical results are calculated from the change in mean cortical thickness in ROIs. As such we prefer to keep this figure in the supplementary material (as it shows individual whole brain data).

Comment-17: - “Longitudinal data: FA and CSA along the spinal cord” sub-section: the results of your previous study should not be placed in the results section but need to be discussed.

Our response: We are not reporting the results from the previous study in this sub-section, but stating that those results are previously reported with citation to the previous paper. Now we have removed this subsection title and merged those sentences into the next subsection (multimodal analysis, where we use the spinal cord measures with the brain measures). We made a similar change in the cross-sectional results section too (removed the subsection title ‘Cross-sectional data: Spinal cord FA and CSA at C2’ and merged it with the next subsection).

Discussion

Comment-18: - Due to your sample size, you need to be more cautious and weight up your arguments when discussing results from a sub-group of your ALS population.

Our response: We have revised the corresponding sentences to convey that our findings are of exploratory nature, especially in the subgroup analyses. We have also added the following paragraph to the Discussion section.

“Our study, like many others, is limited by the relatively small number of participants and high drop-out over the course of 12 months. This remains a shortcoming of ALS imaging studies with relatively long periods of follow-up. Also, the site of onset analysis and other subgroup analyses we conducted have more of an exploratory nature due to the limited number of participants in each subgroup.”

Reviewer #3

General comment: In this study, the authors used a multimodal approach to identify the cortical and spinal cord measures that are altered in ALS. Clinical data (ALSFRS-R, ECAS) were collected, and multimodal MRI was done at baseline and for a subset of participants at longitudinal visits at month 6 and month 12. Neurological examinations were conducted at baseline and 12 months. The study was performed on two scanners: Siemens Trio and Prisma. The authors used brain diffusion and cortical thickness measures to identify alterations in cortical gray and white matter. Additionally, the authors present a relatively novel method of assessment of brain white matter microstructure in ALS using a fixel-based approach. An assessment of MRI measures and their correlations with clinical variables (ALSFRS-R) were performed at baseline and at longitudinal follow up time points of 6 and 12 months.

The main findings were that DTI- and fixel-based measures indicated alterations in white matter microstructure (reduced FA and Fsum) in motor and extra-motor regions in ALS at baseline and longitudinally. There was cortical thinning at baseline in the precentral gyrus. Longitudinal changes in cortical thickness did not survive corrections for multiple comparisons. No significant alterations of brain FDC (combined fiber density and cross-section) were observed at baseline. However, FDC declined longitudinally in white matter tracts subserving the cortical motor regions. Specifically, FDC was declined in slowly progressing patients (functional decline < 0.5 points/month). Significant clinical correlations were found between ALSFRS-R and mean FDC of the right and left corticospinal tract at baseline. No significant correlations were observed between imaging and clinical measures longitudinally. At the 6-month follow up timepoint, marginally significant alterations in fiber density and fiber cross-section values were observed. Predictive analyses were performed at baseline and follow up. These models utilized individual imaging metrics, either individually or as a combination, to predict disease status.

In summary, the authors conclude that, in a slowly progressing ALS cohort, a prediction model with multiple MRI modalities provided the best performance in diagnostic classification cross-sectionally, and that fixel-based measures had the potential to identify longitudinal changes in white matter microstructure.

This manuscript had a number of strengths:

1. The methods were well described and clear.
2. There was a multiregional approach, with MRI of both the brain and spinal cord. The study is relatively unique in this respect.
3. The authors present raw data points, for example, for cross sectional and longitudinal data, as well as for individual subject cortical thickness maps.

4. Multiple modalities were used to assess brain microstructure and cortical thickness to interrogate different aspects of neurodegeneration.

The reviewers propose some major and minor considerations that would be helpful in improving the quality of the manuscript.

Our response: We thank the Reviewer for summarizing our work. Please find below our response and actions as per your suggestions.

Major revisions

Comment-1: - The reviewer would have preferred to see the objectives and hypotheses stated at the end of the Introduction rather than a summary of the methods and findings of the study. If there were no a priori objectives/hypotheses, it would be important to clarify that in the introduction.

Our response: We have carefully revised the introduction section as per this comment and Reviewer 2 comments 2-5. We have removed the mentions of general concepts and issues in dMRI and also removed the description of our work. We now clearly stated our hypotheses and motivations in the introduction.

Comment-2: - It is notable that there is a very large range in disease duration with the inclusion of essentially “long survivors”. Additionally, the disease duration of patients is greater than 3 years on average which is not considered early stage.

a. Thus, there are patients that are progressing much slower than the average person with ALS which can have implications for the ability to detect an effect with these proposed biomarkers; there may have been a longitudinal change observed in the MRI biomarkers ALS patients with a typical rate of functional decline. This should be noted in the Discussion.

b. Are the authors suggesting that within the slowly progressing ALS cohort, there are also early and late stages?

Our response: We mentioned that ours is an early-stage cohort based on the mean ALSFRS-R (40.0). The disease duration varies from 3.4 months to 147.7 months at baseline. The reviewer is right that there are long survivors with very slow progression in our cohort. We have revised this aspect in the Discussion as below. Thanks to the Reviewer for this suggestion.

“Even though we would characterize our cohort as an early-stage cohort based on the mean ALSFRS-R (40.0 at baseline), note that the disease duration varies from 3.4 to 147.7 months, with an average of 38.5 months. This shows that there are long-surviving patients in our cohort who are progressing at a rate much slower than the typical rate of progression in ALS⁴¹. This can have implications for the ability to detect an effect with the proposed biomarkers. This also suggests that these metrics may be highly sensitive to change in a more typical cohort, and possibly over a shorter period of observation. With regard to baseline clinical characteristics such as age (mean 57.5 years), site of onset

(majority limb-onset), and gender (slight majority male), our cohort represents a typical ALS population. The sole atypical feature, slow progression, (ALSFRS-R slope of 0.40 points/month compared to the reported average change of 0.9 to 1 point/month⁴¹), highlights the value of our findings. Despite a small sample size of largely slow progressors, we observed a statistically significant longitudinal change in Fsum and FDC.”

Comment-3: - The ALSFRS-R upper limb “sub-score” is incorrect; the range should be 0-12 not 0-36.

Our response: Thanks to the Reviewer for pointing this out. What we mean here is the ALSFRS-R minus the score for bulbar (questions 4-12). As such we have renamed this sub-score as “ALSFRS-R spinal sub-score (range 0-36)”.

Comment-4: - Page 7, lines 10-11: A suggestion for the authors would be to summarize the findings on spinal cord cross-sectional FA and CSA from their previous publication. This would be helpful to interpret the findings from the current study.

Our response: Based on this comment and other related comments (comments 6 and 17 of Reviewer 2), we have added the below description of our prior work to the Introduction section (we did not include it in the Results section as per the comments we received).

“In our spinal cord MRI paper, we reported the results of a tract-specific and along-the-tract analysis of different spinal cord measures including FA and CSA, from C2 to C6. We demonstrated that the tract-specific analysis with segmentation of ascending and descending tracts in the spinal cord white matter substantially increases the sensitivity of dMRI to disease-related changes in ALS. We identified the tracts and spinal levels affected in ALS, and reported the involvement of sensory pathways in ALS. We also noted strong correlations between spinal diffusion and cross-sectional area metrics and the spinal components of ALSFRS-R.”

Comment-5: - Page 7, lines 22-23: Low UMN burden scores do not automatically indicate lower motor neuron predominance. For patients with low UMN burden scores, it is also suggested that the authors present LMN-specific data. Is there a LMN burden score? Patients with high LMN burden scores and low UMN burden scores would be considered to have LMN-predominant ALS.

Our response: We agree that all patients who have low UMN score cannot be classified as LMN-predominant ALS. We have now computed the LMN burden scores for our ALS participants and identified the LMN predominant participants. We have defined the LMN predominant participants as patients who received low UMN score (1 or 2) and high LMN score. High LMN score is defined as severe weakness (antigravity or worse) in 2 or more limbs. Based on this definition, we have identified 6 of our ALS participants as LMN predominant participants. We conducted the analysis for this subgroup, updated Table 2, marked these participants in Figure 2, and added the below subsection in the results section. We also edited the sentence to which the Reviewer was referring.

“Cross-sectional data: Distinguishing LMN predominant ALS participants from controls

We have defined the LMN predominant participants as participants who received low UMN score (1 or 2) and high LMN score. High LMN score is defined as severe weakness (antigravity or worse) in 2 or more limbs. Six participants were identified in this subgroup (Figure 2). Brain FA, Fsum, and precentral cortical thickness distinguished these participants from controls with high statistical significance (Table 2, last column). However the spinal cord measures did not distinguish the LMN predominant ALS participants from controls. As we only had 6 participants in this category, we have not conducted the accuracy and sensitivity cross-validation analysis for this subgroup.”

We added the below paragraph to the discussion, based on the above result.

“Notably, brain FA, Fsum, and cortical thickness were lower in the 6 LMN-predominant ALS participants than in controls, and this difference was highly significant. The reason for this is unclear. While one could speculate that it supports the view that UMN pathology is the principal driver of disease in ALS, it may simply reflect the fact that severe weakness is a marker of more advanced disease, and that UMN signs are masked by weakness at this point.”

We also added the below sentence to the Methods section.

“A lower motor neuron (LMN) burden score was derived based on the scoring system proposed by Devine et al.⁴⁹. The LMN burden score ranges from 0 to 12, with a higher score indicating greater LMN burden.”

Comment-6: - It is unusual to see an increase in ALSFRS-R after 12 months. This is a reflection of the slowly progressing and unusual ALS cohort in this study which is probably very heterogeneous. Further, for this single individual, could the ALSFRS-R have increased due to symptomatic therapy (for example, of sialorrhea or spasticity)?

Our response: Please refer our response to Reviewer 1, comment 7 (copied below).

We thank the Reviewer for asking about this particular participant. This participant had the same ALSFRS-R at 12 months (a score of 45) as at baseline. As such the change in ALSFRS-R for this subject at 12-months is actually zero. At 6 months, this participant had an ALSFRS-R of 47 which was mistakenly taken as 12-month data. From baseline to 6 months, there was therefore an increase of 2 in ALSFRS-R, which likely reflects test-retest variability. We corrected this error in the paper, and also double checked the ALSFRS-R data for the other participants.

Comment-7: - The sample size in this study is very small. It is suggested that the authors discuss the implications of that in more detail in the discussion in terms of the reliability of identifying biological mechanisms of disease and limitations arising due to small sample sizes.

Our response: We have added the below paragraph in the Discussion.

“Our study, like many others, is limited by the relatively small number of participants and high drop-out over the course of 12 months. This remains a shortcoming of ALS imaging studies with relatively long periods of follow-up. Also, the site of onset analysis and other subgroup analyses we conducted have more of an exploratory nature due to the limited number of participants in each subgroup. Nonetheless, given the consistency of our results across imaging modalities and analysis methods, as well as our thorough statistical analysis, we believe that our findings are reliable and useful to inform us about the pathophysiology of ALS.”

Comment-8: - The authors mention that post hoc analysis of scanner effects across visits did not have a significant effect on the data reported in the study. A description of the analyses performed, and the outcomes should be added to the study.

Our response: It is true that we used two scanners in the study: Siemens Trio and Prisma. However please note that only 4 out of the 24 follow-up scans were conducted on a scanner other than the one used for the baseline scan. Ten of the 20 baseline ALS scans were conducted on Trio and the other 10 ALS scans were conducted on Prisma. For the controls, 7 are scanned on Trio and 13 are scanned on Prisma, at baseline. The analysis we performed is using this data, as detailed below. We have added the below subsection to the manuscript, in the Methods section.

“Tests to study the effect of scanner on MRI metrics

Due to a scanner upgrade during the study, we used both Trio and Prisma scanners for both baseline and follow-up scans. Sequence parameters and protocols were carefully matched. At baseline, 10 of the 20 ALS participants were scanned on Trio and the other 10 were scanned on Prisma. For the controls, 7 participants were scanned on Trio and 13 participants were scanned on Prisma at baseline. We conducted two-sample, two-tailed, unpaired t-tests between the two scanner groups in the control participants at baseline, and did not find statistically significant differences in any of the metrics we used. We also conducted a similar analysis between the two scanner groups in ALS participants, which too did not detect statistically significant differences.”

Comment-9: - It is the preference of the reviewers that discussion points should not be included in the results. Therefore, it is suggested that the authors' qualification of the Fsum findings in the results should be moved to the Discussion section.

Our response: It is not clear which statement/statements the reviewer is referring to here. We have moved the below sentence that compares Fsum and FA to the discussion.

“Fsum detected baseline differences between ALS and control participants in the largest number of brain regions, but FA provided higher effect size and higher statistical significance (lower p-value). This may reflect the lower variance in FA values.”

Comment-10: - The 95% confidence intervals appear to be inaccurate in figure 2. It is recommended that the authors re-generate these plots with accurate estimates of confidence intervals.

Our response: We are unsure what aspect of the intervals displayed in the figure gives the impression of inaccuracy. We have confirmed that the 95% confidence interval (CI) calculations in Figure 2 are correct, using both Prism and MATLAB software.

Comment-11: - Clinical correlations: As the ALSFRS-R score is largely driven by the impairment of lower motor neurons, it is suggested that the authors report correlations between brain metrics and clinical measures pertaining to upper motor neurons, for example, with UMN burden scores.

Our response: We appreciate this important point, we conducted this analysis, and have added the following sentence:

“There were no statistically significant correlations between any of the MRI measures and UMN burden score.”

Comment-12: - It is commendable that the authors present individual subject data on longitudinal cortical thickness measurements in the supplementary material. Although cortical thickness measurements show a marginally significant reduction over a 1 year follow up in ALS, the figures in the supplementary section indicate that there is cortical thickening over time. We suggest that the authors clarify the discordance between actual cortical thickness alterations and their interpretation of these alterations. It is also suggested that the authors consider presenting increases/decreases in cortical thickness as separate figures.

Our response: This supplementary figure mostly shows cortical thinning. We have modified this figure by updating the color scale used, to avoid the ambiguity that was present in using light blue for cortical thinning and dark blue for cortical thickening. The reviewer is still right that this figure shows cortical thickening (10-15%) in some regions in some of the participants. We explained this in the discussion section.

“We noted a large variability in individual cortical thickness in some regions, especially in frontal and temporal poles, including an increase in cortical thickness in these regions in some participants (Supplementary figure 2). This variability is likely due to the small size and the location of these regions which makes it challenging for the freesurfer to segment them accurately. This increase was not statistically significant, however.”

Minor revisions

Comment-13: - The authors mention the use of a ball-and-stick model (TBSS) as compared to the tensor model (DTI) (page 3, line 19). It would be helpful for the authors to clarify the rationale for the choice of a TBSS over DTI model. A short description of the novel fixel-based measures and their relevance to health and disease would be useful.

Our response: We have used both DTI and ball-and-stick model in the TBSS analysis and compared both sets of results. The benefit of the ball-and-stick model is its capability to resolve crossing fibers. We have clarified this in the Introduction section (see below) and also discussed the comparisons (Fsum Vs. FA) in the Discussion section.

“We use the *ball-and-stick* multi-compartment model that resolves crossing fibers (FSL *bedpostx* algorithm²⁸) in a TBSS analysis rather than using the DTI model. The *ball-and-stick* model uses a ball compartment to represent the isotropic diffusion and multiple stick components to represent the anisotropic diffusion in crossing fibers.”

We have also added the below new subsection about the fixel-based measures in the methods section.

“Fixel-based Analysis

The fixel-based analysis (FBA) addresses the challenge of resolving crossing fibers by extracting measures that are linked to individual fiber populations (called *fixels*) within a voxel that are directly related to the white matter anatomy. The amplitudes of the fiber orientation distributions (FOD) were calculated to derive the fixels, using constrained spherical deconvolution⁵⁴. The FODs were segmented to evaluate the orientation and number of fixels in each voxel and generate the fiber density (FD), fiber-bundle cross-section (FC) and a product of both, fiber density and cross-section (FDC) maps. The identification of related fixels is driven by a template tractogram generated with more anatomically relevant white matter tracts⁵⁵. FBA metrics describe the total intra-axonal volume that is related to the white matter microstructure; and the fiber bundle cross-section that describes the macroscopic differences in fiber bundle. Due to their specificity, FBA metrics are able to differentiate between affected and non-affected tracts within a given voxel unlike the conventional DTI that resolves only one fiber population⁵⁶. Therefore, FBA is more useful in characterizing the complex geometry of the white matter with more clinical relevance pertaining to its biological specificity.”

Comment-14: - It is very interesting to note that none of the patients in King’s stage 3/4 at baseline returned for a follow up scan at 12 months. It would be important to see how the patients at King’s stage 1/2 are different from patients at stage 3/4, in terms of MRI metrics and demographic characteristics at baseline. However, it is accepted that this would be an exploratory analysis given the small sample size.

Our response: Thank you for this suggestion. We did this analysis and found marginally significant differences in Fsum and FA between King’s stage 1/2 and 3/4 participants. We have added this result to the paper, as below.

“Cross-sectional data: Analysis based on King’s staging

We conducted an exploratory analysis comparing the MRI measures of participants who were at King’s stage 3 or 4 at baseline with those measures of participants at King’s stage 1 or 2. We found marginally significant (that is p-values were not significant after correction for multiple testing) differences in Fsum and FA between these participants. Both Fsum (uncorrected p-value: 0.022) and FA (uncorrected p-value: 0.025) were lower in participants at King’s stage 3 or 4 compared to participants at King’s stage 1 or 2.”

Comment-15: - In addition to the exploratory analyses presented of spinal vs bulbar ALS, it would be interesting to explore disease heterogeneity with a comparison of “fast” and “slow” progressors.

Our response: We have now conducted this analysis. We compared the baseline data of 6 ALS subjects who had a change in ALSFRS-R of <0.5 points/ month with that of 5 ALS subjects who had a change in ALSFRS-R of ≥ 0.5 points/ month (and less than 1.0 points/ month since we did not have any participants with 1 or more points change per month). We did not find statistically significant differences between these two groups. We also studied the difference in 12-month longitudinal change in these participants and did not find statistically significant differences. Note that we had a total of 11 participants only for this experiment which are divided into subgroups, compared to 20 in the other subgroup analysis on the cross-sectional data to which the reviewer is referring. As such, we have not included this result in the manuscript.

Comment-16: - Was a cutoff employed for the ECAS total score to select healthy participants for inclusion in the study?

Our response: Yes, we used a cutoff of 105 in total ECAS score to select healthy participants. However we still included one control participant who had a total ECAS score 103 in the cohort as the participant was cognitively normal otherwise and as the participant was tested at 6-months and 12-months with ECAS scores 125 and 123 respectively. We have added the following text to the manuscript to convey this.

“One control participant scored (ECAS score 103) slightly below the cognitive screen normal cutoff (ECAS score 105) at enrollment but was included as this was determined to be due to a misunderstanding of one of the instructions; cognitive scores at 6 and 12 months were well within normal limits for this participant (ECAS scores of 125 and 123, respectively).”

---- END ----

REVIEWERS' COMMENTS:

Reviewer #1 (Remarks to the Author):

Authors have implemented a discrete number of changes and have substantially revised the entire manuscript. They have included additional analysis based on LMN score, King's staging, inclusion of 6-months clinical and demographic data and have majorly revised both introduction and discussion sections, making the manuscript clearer and more complete.

- One of the major concerns was that part of the results was already published by the same authors. However, they have clarified that only cohort and cohort characteristics have remained the same, but the analysis provided here also included brain data and not only spinal cord data. This clarifies the first concern.
- Sample size remains quite small. However, authors have discussed this limitation in the appropriate section.
- Authors have adequately justified the analysis of data regarding only the C2 spinal level and added a satisfactory comment in the discussion section
- Analysis of MRI metrics across patients with upper and lower UMN burden has been performed and commented.
- Statement on how brain FA and brain Fsum have been calculated has been made explicit both in the results and methods section.
- P values on longitudinal data have been added.
- Clarification and correction on the one subject's ALS-FRS-R has been performed.
- Genetic analysis has been made explicit.
- Table 1 quality has been improved.

Authors have adequately implemented all suggestions in the current version of the manuscript and have modified it accordingly.

Two very minor imprecisions are below:

- Line 70: "CSA" please specify the abbreviation as it is the first time it appears in the main text
- Figure 2-B: please notice that "Fsum" results underlined and modified it accordingly.

Other than these two minor revisions, I shall have no further concerns for the publication of the paper.

Reviewer #2 (Remarks to the Author):

I would like to thank the authors for their important contribution to understand a challenging disease for diagnosis, patients stratification and prognosis.

Improved Diagnostic Accuracy and Sensitivity to Longitudinal Change in ALS with Brain and Cervical Cord MRI

Response to review comments

We thank the Reviewers for re-reviewing our work and for acknowledging the importance of the contributions of our work. We have revised the paper as per the minor comments we received (specifying the abbreviation for CSA as cross-sectional area and modifying Figure 2b to remove the red underline) and also as per the editor's comments in the checklist we received.